# Short-term social isolation acts on hypothalamic neurons to promote social behavior in a sex- and context-dependent manner

Xin Zhao, Yurim Chae, Destiny Smith, Valerie Chen, Dylan DeFelipe, Joshua W Sokol, Archana Sadangi, Katherine Tschida*

Department of Psychology, Cornell University, Ithaca, United States

**\*For correspondence:**
kat227@cornell.edu

**Competing interest:** The authors declare that no competing interests exist.

## eLife Assessment

This **important** study substantially advances our understanding of the neural circuits that regulate social behavior by identifying a population of hypothalamic neurons in the preoptic area that promote social interactions following short-term isolation. The evidence supporting the authors' claims is **solid**, with well-designed experiments using validated activity-dependent tagging and manipulation methods, though some differences in outcomes between experiments highlight limitations of the tagging approach. The work will be of broad interest to neuroscientists studying social behavior, neural circuit function, and hypothalamic mechanisms and will represent a meaningful contribution to the field.

**Abstract** Social animals, including both humans and mice, are highly motivated to engage in social interactions. Short-term social isolation promotes social behavior, but the neural circuits through which it does so remain incompletely understood. Here, we sought to identify neurons that promote social behavior in single-housed female mice, which exhibit increased rates of social investigation, social ultrasonic vocalizations (USVs), and mounting during same-sex interactions that follow a period of short-term (3 days) isolation. We first used immunostaining for the immediate early gene Fos to identify a population of neurons in the preoptic hypothalamus (POA) that increase their activity in single-housed females following same-sex interactions (POA$_{social}$ neurons) but not in single-housed females that did not engage in social interactions. TRAP2-mediated chemogenetic silencing of POA$_{social}$ neurons in single-housed females significantly attenuates the effects of short-term isolation on social investigation, USV production, and mounting. In contrast, caspase-mediated ablation of POA$_{social}$ neurons in single-housed females robustly attenuates mounting but does not decrease social investigation or USV production. Optogenetic activation of POA$_{social}$ neurons in group-housed females promotes social investigation and USV production but does not recapitulate the effects of short-term isolation on mounting. To understand whether a similar population of POA$_{social}$ neurons promotes social behavior in single-housed males, we performed Fos immunostaining in single-housed males following either same-sex or opposite-sex social interactions. These experiments revealed a population of POA neurons that increase Fos expression in single-housed males following opposite-sex, but not same-sex, interactions. Chemogenetic silencing of POA$_{social}$ neurons in single-housed males during interactions with females reduces mounting but does not affect social investigation or USV production. These experiments identify a population of hypothalamic neurons that promote social behavior following short-term isolation in a sex- and social context-dependent manner.

## Introduction

Humans and other social mammals find social interactions rewarding and are highly motivated to seek out social connections. Consequently, the experience of social isolation is aversive and impacts both our brains and our behaviors. While long-term isolation can lead to the emergence of anti-social behaviors in both humans and rodents (*An et al., 2017*; *Arrigo and Bullock, 2008*; *Check et al., 1985*; *Hossain et al., 2020*; *Killgore et al., 2021*; *Ma et al., 2011*; *Ma et al., 2022*; *Machimbarrena et al., 2019*; *Matsumoto et al., 2005*; *Mears and Bales, 2009*; *Reid et al., 2022*; *Toth et al., 2011*; *Valzelli, 1973*; *Weiss et al., 2004*; *Wiberg and Grice, 1963*; *Zelikowsky et al., 2018*), short-term isolation typically increases levels of social motivation and promotes social-seeking behaviors (*Baumeister and Leary, 1995*; *Cacioppo et al., 2006*; *Cacioppo and Cacioppo, 2018*; *House et al., 1988*; *Lee et al., 2021*; *Niesink and van Ree, 1982*; *Panksepp and Beatty, 1980*; *Zhao et al., 2021*). Alterations in social behavior are characteristic of many neurodevelopmental disorders, including autism spectrum disorder (*Chevallier et al., 2012*; *Clements et al., 2018*). How short-term social isolation acts on the brain to promote social behavior remains incompletely understood.

The mesolimbic pathway (i.e. projections from dopaminergic ventral tegmental neurons to the ventral striatum) plays an important role in regulating social motivation and social reward, during courtship as well as during same-sex interactions (*Bariselli et al., 2018*; *Dai et al., 2022*; *Dölen et al., 2013*; *Gunaydin et al., 2014*; *Hung et al., 2017*; *Love, 2014*; *Melis et al., 2022*; *Resendez et al., 2020*; *Robinson et al., 2011*; *Solié et al., 2022*; *Tang et al., 2014*; *Walum and Young, 2018*; *Xiao et al., 2017*). In line with its role in regulating social behavior, changes in the function of the meso-limbic pathway have been reported following long-term (weeks-long) social isolation and/or early-life social isolation (*McWain et al., 2022*; *Musardo et al., 2022*; *Tan et al., 2021*; *Yorgason et al., 2016*). The neural circuit changes that mediate the effects of short-term isolation on social behavior in adult animals are comparatively less explored, but here as well, recent studies in both humans and rodents have implicated changes in various populations of midbrain dopamine neurons (*Inagaki et al., 2016*; *Matthews et al., 2016*; *Tomova et al., 2020*). Beyond its effects on mesolimbic circuits, whether social isolation acts on additional neuronal populations to promote social behavior is unknown.

In recent work, we found that short-term (3 days) social isolation exerts robust effects on the social behaviors of C57BL/6 J female mice (*Zhao et al., 2021*). Relative to group-housed females, single-housed females that subsequently engaged in same-sex interactions exhibited increased rates of social investigation, increased rates of USVs, and were also observed to mount female social partners, a behavior not observed in pairs of group-housed females (*Zhao et al., 2021*). The robust effect of short-term isolation on these three aspects of female social behavior provides a powerful paradigm to identify neurobiological changes that mediate the effects of short-term isolation on social behavior. In the current study, we combined this behavioral paradigm with Fos immunostaining and the TRAP2 activity-dependent labeling approach (*Allen et al., 2017*; *DeNardo et al., 2019*) to identify and characterize a population of neurons in the preoptic hypothalamus that increase their activity in single-housed females following same-sex social interactions (i.e. POA$_{social}$ neurons). We next asked whether silencing or ablation of POA$_{social}$ neurons in single-housed females attenuates female social behaviors, and whether artificial activation of POA$_{social}$ neurons in group-housed females promotes social behaviors. Finally, we extended a subset of these experiments to single-housed males engaged in opposite-sex and same-sex interactions, to understand whether short-term isolation acts on the POA to promote social behavior in a manner that depends on either sex or social context. This study identifies novel neurobiological mechanisms through which short-term social isolation acts on the brain to promote social interaction. Our findings also add to an emerging literature indicating that the POA regulates not only sexual behavior but also female social behavior during same-sex interactions.

## Results

### Neurons in the preoptic hypothalamus increase Fos expression in socially isolated female mice following same-sex social interactions

To identify changes in neuronal activity that may underlie the effects of short-term isolation on female social behavior, we performed immunostaining for the immediate early gene Fos in brain sections collected from group-housed and single-housed (3 days) subject females following 30-min social encounters in their home cages with a novel, group-housed visitor female (*Figure 1A*). In line with our

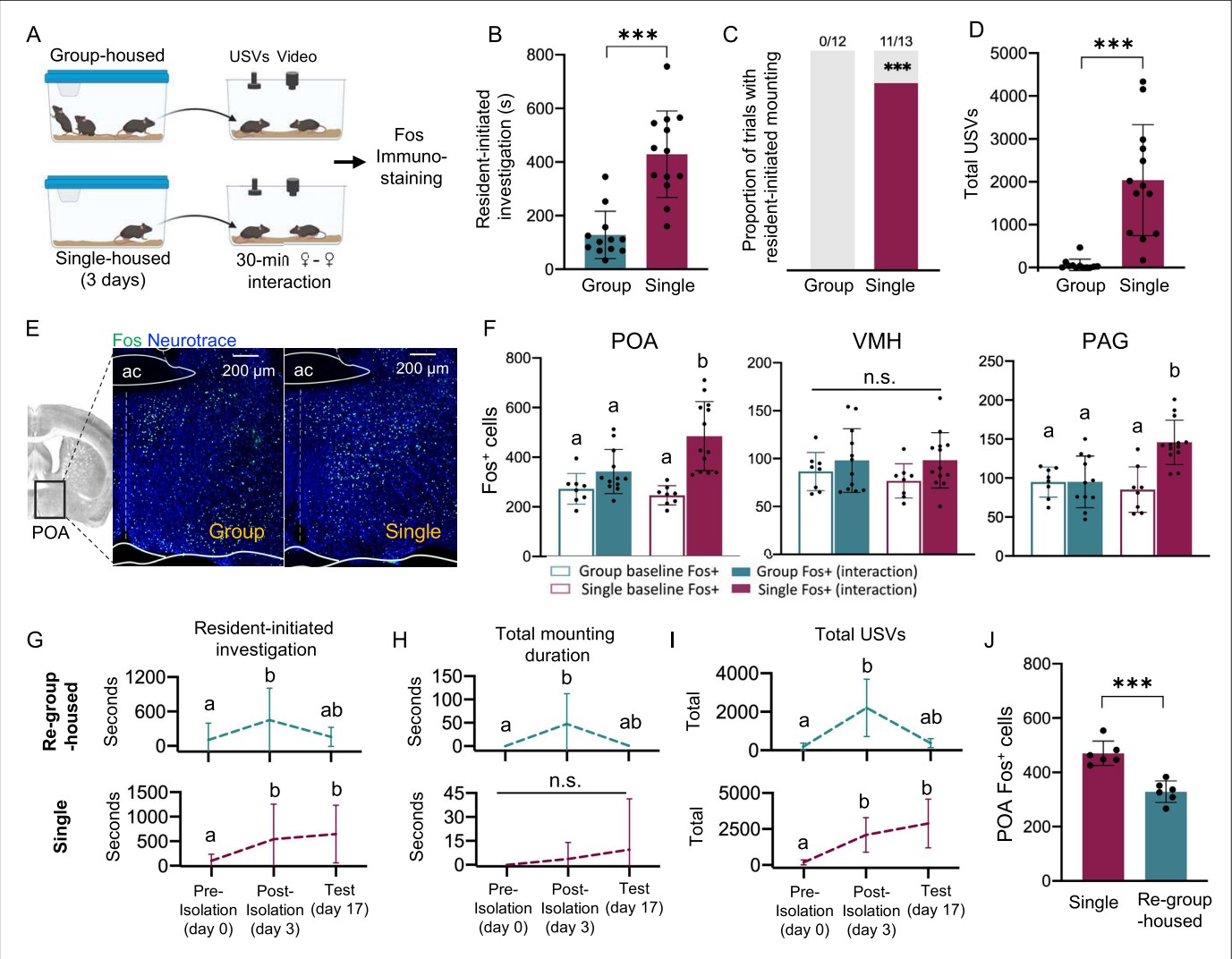

**Figure 1.** The POA contains neurons that increase Fos expression in single-housed females following same-sex interactions. (**A**) Schematic of experiment to measure Fos expression in group-housed and single-housed females following same-sex social interactions. (**B**) Total time spent engaged in resident-initiated social investigation for group-housed residents (teal) and single-housed residents (maroon). (**C**) Same as (**B**), for proportion of trials with resident-initiated mounting. (**D**) Same as (**B**), for total USVs recorded from pairs containing group-housed or single-housed residents. (**E**) Left-most image shows the location of the POA in a coronal brain section. Representative confocal images show Fos expression (green) in the POA of a group-housed female (left) and a single-housed female (right) following same-sex social interactions. Blue, Neurotrace. (**F**) Quantification of Fos-positive neurons is shown for the POA (left), the VMH (middle), and the caudolateral PAG (right) for group-housed and single-housed females. Open bars show data from females that did not engage in social interactions with novel females (baseline), and closed bars show data from females following social interactions with novel females (interaction). (**G**) Total time spent in resident-initiated interaction is plotted for 17 day single-housed (maroon) and re-group-housed females (teal) during same-sex interactions that occurred prior to isolation (day 0), following 3 days of isolation (day 3), and on the test day (day 17). (**H**) Same as (**G**), for total resident-initiated mounting time. (**I**) Same as (**G**), for total USVs. (**J**) Quantification of Fos-positive POA neurons is shown for 17 day single-housed females (maroon) and re-group-housed females (teal). All error bars show standard deviation.

The online version of this article includes the following figure supplement(s) for figure 1:

**Figure supplement 1.** Additional characterization of Fos expression in single-housed vs. group-housed females and comparison to rates of female social behaviors.

previous behavioral findings (*Zhao et al., 2021*), we observed that single-housed female residents spent more time investigating visitors (*Figure 1B*; t-test, p=0.001) and in many trials mounted visitors, a behavior that was not observed in group-housed residents (*Figure 1C*; 0 of 12 group-housed residents and 11 of 13 single-housed residents mounted visitors; z-test for independent proportions, p<0.001; see *Supplementary file 1* for complete statistical details). Female pairs that contained a

single-housed resident also produced higher rates of ultrasonic vocalizations (USVs) than pairs with a group-housed resident (*Figure 1D*; Mann-Whitney U test, p<0.001). Although either female in a dyad can produce USVs (*Warren et al., 2020*), the robust effects of short-term isolation on the non-vocal social behaviors of single-housed females suggest that at least some of the elevation in total USVs is driven by increased USV production by the single-housed resident. Given the robust effects of short-term isolation on these three aspects of female social behavior, we focused our analyses on two hypothalamic regions implicated in regulating these behaviors: the preoptic area (POA), which regulates social approach (*McHenry et al., 2017*), social reward (*Hu et al., 2021*), mounting (*Floody, 1989*; *Karigo et al., 2021*; *Wei et al., 2018*) and USV production (*Chen et al., 2021*; *Gao et al., 2019*; *Green et al., 2018*; *Karigo et al., 2021*; *Michael et al., 2020*); and the ventromedial hypothalamus (VMH), which regulates mounting (*Hashikawa et al., 2017*; *Karigo et al., 2021*; *Lee et al., 2014*; *Liu et al., 2022*). We also examined Fos expression within the caudolateral periaqueductal gray (PAG), based on the well-established role of this region in the control of vocalization in vertebrates and USV production in mice (*Chen et al., 2021*; *Jürgens, 1994*; *Michael et al., 2020*; *Tschida et al., 2019*; *Ziobro et al., 2024*). To test whether any observed differences in Fos expression in these three regions were associated with isolation-induced changes in social behavior rather than baseline differences between groups, we also measured Fos expression in the POA, the VMH, and the PAG of group-housed and single-housed females that did not engage in social interaction with novel female visitors (*Figure 1E and F*; *Figure 1—figure supplement 1*).

These analyses revealed that baseline levels of Fos expression within the POA and the VMH did not differ between group-housed and single-housed females (*Figure 1F*, left and middle, open bars; two-way ANOVA to analyze Fos expression within each brain region, factor 1=housing status, factor 2=social interaction, followed by post-hoc Tukey's HSD tests). Following social interactions with novel female visitors, single-housed females exhibited robust increases in Fos expression within the POA (*Figure 1F*, left; p<0.001) but not within the VMH (*Figure 1F*, middle; p>0.05). In contrast, Fos expression within these two brain areas did not increase significantly in group-housed females that interacted with novel female visitors (*Figure 1F*; p>0.05 for both comparisons). Similar to the POA, baseline Fos expression within the PAG did not differ between group-housed and single-housed females (p>0.05), and only single-housed females displayed increased PAG Fos expression following social interactions with novel female visitors (*Figure 1F*, right; p<0.001), a finding that further supports the idea that single-housed females increase USV production during same-sex interactions. POA Fos expression was significantly and positively correlated with the total amount of time spent in resident-initiated investigation for both group-housed and single-housed females (*Figure 1—figure supplement 1C*; linear regression, p<0.05), as well as with the total time spent mounting by single-housed resident females (*Figure 1—figure supplement 1C*; p=0.01). In both group-housed and single-housed female residents, POA Fos expression tended to correlate positively with total USVs, but these relationships were not significant (*Figure 1—figure supplement 1C*; p=0.05 for group-housed and p=0.09 for single-housed; see *Figure 1—figure supplement 1D, E* also for relationships of VMH Fos and PAG Fos to vocal and non-vocal social behaviors). In summary, POA Fos expression increases selectively in single-housed females following same-sex social interactions, and levels of POA Fos expression are also well related to the production of specific types of social behaviors by both group-housed and single-housed females.

To ask whether the effects of short-term isolation on female social behavior and POA Fos expression are long-lasting, we measured social behaviors of female residents at three timepoints: (1) on day 0, when female subjects were still group-housed; (2) on day 3, after female subjects had been single-housed for 3 days; and (3) on day 17, after half of the subject females had been re-group-housed with their same-sex siblings for two weeks and the other half of the subject females remained single-housed for an additional two weeks (*Figure 1G–I*). Brains of re-group-housed and 17 day single-housed subject females were collected 2 hr after the start of the day 17 social interaction, and Fos expression within the POA was examined (*Figure 1J*). Consistent with our earlier findings, rates of social investigation and USV production significantly increased following 3 days of social isolation (*Figure 1G and I*; p<0.05 for day 0 vs. day 3 in both groups for both behaviors). Following re-group-housing, time spent in social investigation and rates of USV production tended to decrease to pre-isolation levels (*Figure 1G and I*, top plots; p=0.08 for day 0 vs. day 17 investigation time and p=0.06 for day 0 vs. day 17 total USVs in re-group-housed females). In contrast, females that

were single-housed for 17 days continued to spend increased time in social investigation (*Figure 1G*, bottom plot; p<0.05 for day 0 vs. day 3 investigation and for day 0 vs. day 17 investigation), and pairs containing 17-day single-housed residents continued to produce elevated rates of USVs (*Figure 1I*, bottom plot; p<0.05 for day 0 vs. day 3 USVs and for day 0 vs. day 17 USVs). Time spent mounting tended to follow the same trends as rates of social investigation and USV production in re-group-housed and 17-day single-housed females (*Figure 1H*). Along with the attenuation of female social behaviors following re-group-housing, we also found that POA Fos expression was significantly lower in re-group-housed females relative to 17 day single-housed females (*Figure 1J*; t-test, p<0.001). These findings support the idea that changes in female social behavior following short-term isolation are reversible and are accompanied by decreased POA Fos expression. Hereinafter, we refer to the population of POA neurons that increase Fos expression in single-housed females that have engaged in same-sex interactions as POA$_{social}$ neurons, and we next conducted experiments to test whether functional manipulations of POA$_{social}$ neuronal activity impact the effects of short-term isolation on female social behavior.

## Chemogenetic inhibition of POA$_{social}$ neurons attenuates social investigation, mounting, and USV production in single-housed females

If increased activity of POA$_{social}$ neurons promotes social behavior in single-housed females, one prediction is that reducing the activity of POA$_{social}$ neurons in single-housed females will attenuate the effects of isolation on female social behavior. To test this idea, we employed the TRAP2 activity-dependent labeling strategy to chemogenetically silence POA$_{social}$ neurons in single-housed females during social interactions with novel, group-housed female visitors (*Figure 2A*). Briefly, the POA of TRAP2 female mice was injected bilaterally with a virus driving the Cre-dependent expression of the inhibitory DREADDs receptor hM4Di. Three weeks later, females were single-housed for 3 days and then given a 30-min social encounter with a novel, group-housed female visitor in their home cage. Following the social interaction, resident females were given an I.P. injection of 4-hydroxytamoxifen (4-OHT), which drives the transient expression of Cre recombinase in recently active neurons and thereby enables the expression of hM4Di in POA$_{social}$ neurons. Subject females remained single-housed for an additional 24 hr and then were re-group-housed with siblings for 2 weeks. Subject females were then single-housed a second time for 3 days and subsequently given a 30-min same-sex interaction following I.P. injection of either saline (control) or clozapine-n-oxide (CNO) (saline and CNO tests were run 3 days apart, females remained single-housed during these 3 days, and the order was counterbalanced across experiments).

Comparison of the social behaviors of single-housed females between CNO and saline sessions revealed that chemogenetic silencing of POA$_{social}$ neurons significantly reduced resident-initiated investigation (*Figure 2B*; n=17; red points; two-way ANOVA with repeated measures on one factor; p<0.01). Inhibition of POA$_{social}$ neurons also significantly reduced the proportion of females that exhibited mounting (*Figure 2C*; McNemar's test for paired proportions; p=0.04 for POA$_{social}$ CNO vs. saline). Finally, inhibition of POA$_{social}$ neurons significantly reduced USV production (*Figure 2D*; two-way ANOVA with repeated measures on one factor; p<0.01). In contrast, CNO treatment did not affect the production of any of these social behaviors in single-housed females with GFP expressed in POA$_{social}$ neurons (*Figure 2B–D*; n=14; black points; p>0.05 for all CNO vs. saline comparisons in the POA$_{social}$-GFP control group). To investigate the specificity of these effects to chemogenetic silencing of POA$_{social}$ neurons, we also performed control experiments in which activity-dependent chemogenetic silencing in single-housed females was performed caudal to the POA within the anterior hypothalamus (AH; *Figure 2B–D*; n=12; brown points) or within the VMH (*Figure 2B–D*; n=5; gray points). No significant effects of CNO treatment on resident-initiated investigation, mounting, or total USVs were observed in these control groups (*Figure 2C and D*; p>0.05 for all). The effect of chemogenetic inhibition of POA$_{social}$ neurons to decrease female social behavior also cannot be attributed to an overall decrease in movement (*Figure 2E*; paired t-test; p>0.5 for difference in movement between saline and CNO sessions; see Materials and methods).

We next asked whether the magnitude of the effects of chemogenetic silencing of POA$_{social}$ neurons on different female social behaviors was related to the rates at which these behaviors were produced during TRAPing sessions that immediately preceded 4-OHT treatment. To this end, we compared total resident-initiated investigation, total resident-initiated mounting, and total USVs produced

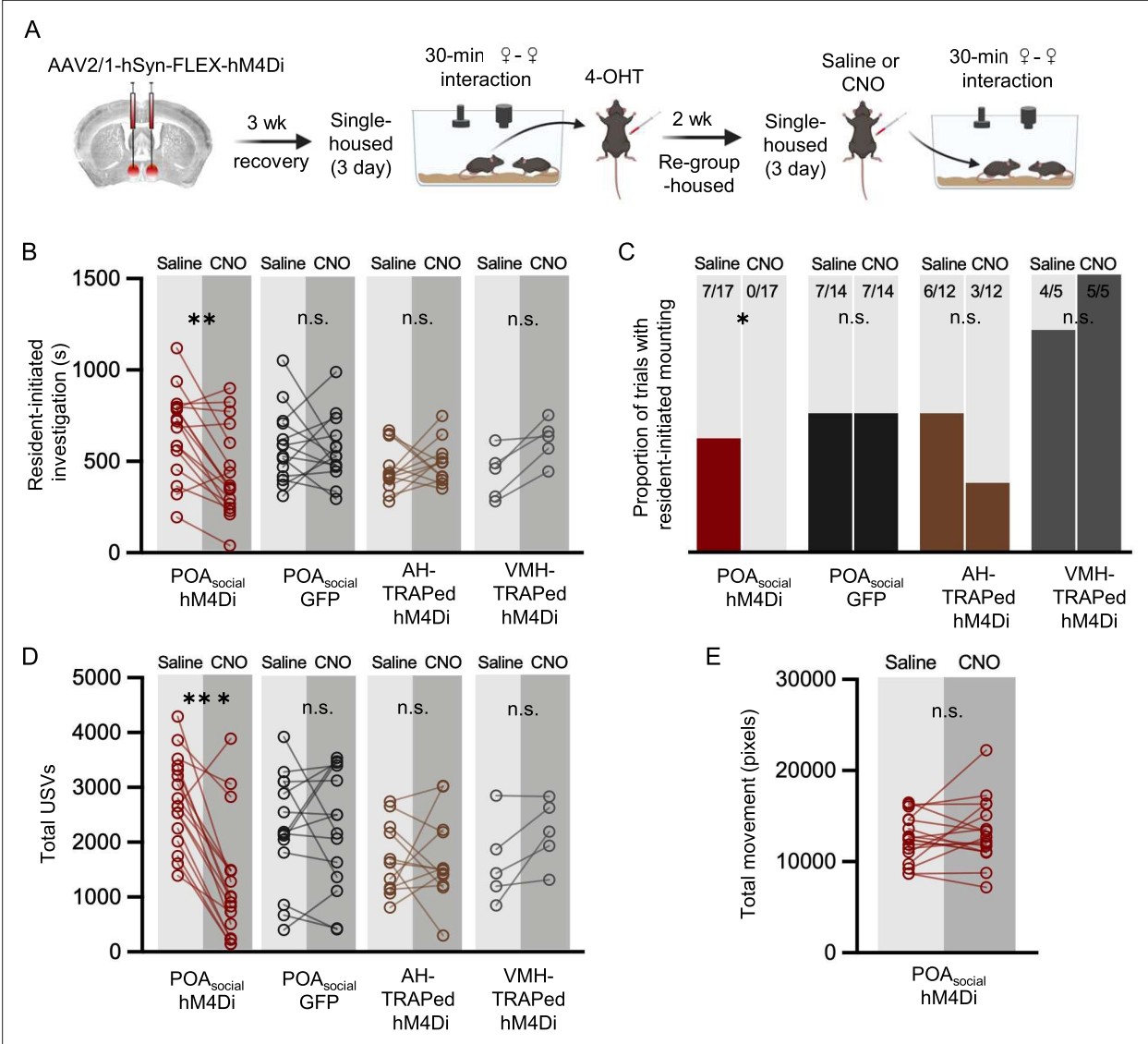

**Figure 2.** Effects of chemogenetic inhibition of POA_social neurons on the social behaviors of single-housed female mice. (**A**) Experimental timeline and viral strategy to chemogenetically inhibit the activity of POA_social neurons in single-housed females. (**B**) Total time spent in resident-initiated social investigation is shown on saline and CNO days for 4 experimental groups: (red symbols) experimental females in which hM4Di is expressed in POA_social neurons; (black symbols) control females in which GFP is expressed in POA_social neurons; (brown symbols) control females in which hM4Di is expressed in 'TRAPed' AH neurons; (gray symbols) control females in which hM4Di is expressed in 'TRAPed' VMH neurons. (**C**) Same as (**B**), for proportion of trials with resident-initiated mounting. (**D**) Same as (**B**), for total USVs. (**E**) Total movement is plotted for females with hM4Di expressed in POA_social neurons, on saline days vs. CNO days.

The online version of this article includes the following figure supplement(s) for figure 2:

**Figure supplement 1.** Additional analyses and control experiments related to **Figure 2**.

during TRAPing sessions to changes in these behaviors for each female following chemogenetic silencing of POA_social neurons (calculated as [CNO behavior − saline behavior]). These comparisons revealed no significant relationships (**Figure 2—figure supplement 1A–C**; linear regression; p>0.05 for all). Although we designed our experiments to TRAP and subsequently manipulate the activity of POA neurons that upregulated Fos expression following the production of social behaviors in single-housed females, one possibility is that our strategy inadvertently led to the labeling of POA neurons that upregulate Fos in a manner that reflects the experience of social isolation per se. To test this idea, we performed chemogenetic silencing of POA neurons in females that were single-housed but were not given a social interaction prior to 4-OHT treatment (n=5 non-social controls; **Figure 2—figure**

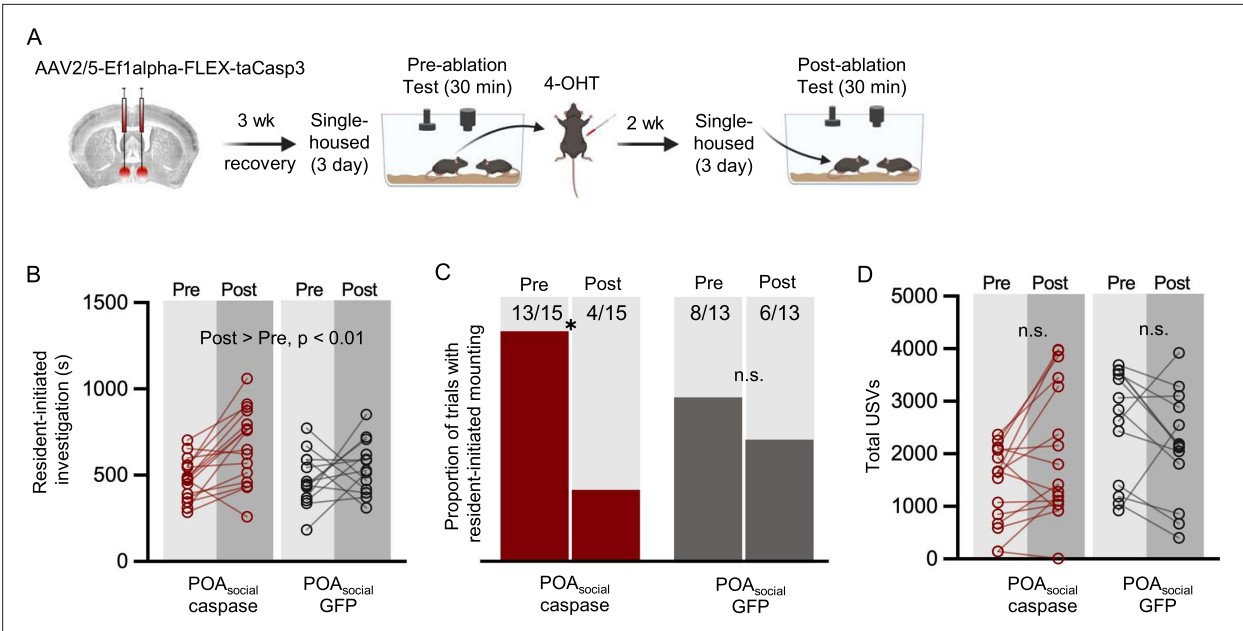

**Figure 3.** Effects of caspase-mediated ablation of POA$_{social}$ neurons on the social behaviors of single-housed female mice. (**A**) Experimental timeline and viral strategy for caspase-mediated ablation of POA$_{social}$ neurons in single-housed females. (**B**) Total time spent in resident-initiated social investigation is shown pre- and post-4-OHT treatment for 2 experimental groups: (red symbols) experimental females in which caspase is expressed in POA$_{social}$ neurons; (black symbols) control females in which GFP is expressed in POA$_{social}$ neurons. (**C**) Same as (**B**), for proportion of trials with resident-initiated mounting. (**D**) Same as (**B**), for total USVs.

The online version of this article includes the following figure supplement(s) for figure 3:

**Figure supplement 1.** Comparison of effects of ablation of POA$_{social}$ neurons in TRAP2 heterozygous vs. homozygous females.

---

*supplement 1D*). In these control single-housed females, CNO treatment had no effect on social investigation, mounting, or USV production during same-sex interactions (*Figure 2—figure supplement 1E–G*; p>0.05). In summary, we demonstrate that chemogenetic silencing of POA$_{social}$ neurons reduces social investigation, mounting, and USV production in single-housed female mice, and that these effects in turn require the production of social behaviors during the TRAPing sessions that precede 4-OHT treatment.

## Ablation of POA$_{social}$ neurons attenuates mounting in single-housed females

In previous work investigating the role of the POA in regulating rodent social behaviors, studies have reported different effects on behaviors according to whether they employed reversible or irreversible neuronal silencing strategies. Studies that used chemogenetic or optogenetic methods to reversibly silence genetically-defined subsets of POA neurons report decreases in both USV production in males (*Chen et al., 2021*; *Karigo et al., 2021*) and mounting in males and females during interactions with female social partners (*Gao et al., 2019*; *Karigo et al., 2021*). In contrast, studies employing caspase-mediated ablation of genetically-defined subsets of POA neurons (*Gao et al., 2019*; *Wei et al., 2018*) or electrolytic lesions of the POA (*Bean et al., 1981*) report decreased mounting but no effects on rates of USV production. To test whether permanent ablation of POA$_{social}$ neurons attenuates social behaviors in single-housed females in a manner similar to the effects of chemogenetic inhibition, we used the TRAP2 activity-dependent labeling strategy to express caspase in and to thereby ablate POA$_{social}$ neurons (*Figure 3A*; see Materials and methods). Vocal and non-vocal social behaviors of resident females were compared pre- and post-ablation, and the same measurements were made in control females expressing GFP in POA$_{social}$ neurons.

Similar to the effects of chemogenetically inactivating POA$_{social}$ neurons, ablation of POA$_{social}$ neurons significantly reduced mounting in single-housed females (*Figure 3C*; McNemar's test for paired proportions; p=0.03 for proportion of trials with mounting on pre-4-OHT vs. post-4-OHT tests in

POA$_{social}$-caspase females; p>0.05 for pre-4-OHT vs. post-4-OHT in POA$_{social}$-GFP females). In contrast to the effects of chemogenetic inhibition of POA$_{social}$ neurons, we found that caspase-mediated ablation of POA$_{social}$ neurons did not affect rates of social investigation in single-housed females, although both experimental and control females spent more time investigating visitors in the post-4-OHT session (*Figure 3B*; two-way ANOVA with repeated measures on one factor; p>0.05 for main effect of group, p<0.01 for main effect of time, p>0.05 for interaction effect). Ablation of POA$_{social}$ neurons also failed to reduce USV production in pairs containing single-housed females (*Figure 3D*; two-way ANOVA with repeated measures on one factor; p>0.05 for pre-4-OHT vs. post-4-OHT total USVs in POA$_{social}$-caspase females). Together with our chemogenetic inhibition data, these results show that both reversible inhibition or irreversible ablation of POA$_{social}$ neurons in single-housed female mice reduces mounting, whereas only chemogenetic inhibition of POA$_{social}$ neurons attenuates the effects of short-term isolation on female social investigation and USV production.

One possibility is that a difference in the efficacy of the strategies used to irreversibly ablate vs. reversibly inhibit female POA$_{social}$ neurons accounts for the difference in their effects on female social behaviors. Indeed, although we used TRAP2 homozygous females in the POA$_{social}$-hM4Di dataset (*Figure 2*), the majority of the females used in the POA$_{social}$-caspase dataset were TRAP2 heterozygotes (11 of 15 females were generated by crossing TRAP2 mice with Ai14 mice, and 4 of 15 were TRAP2 homozygotes; absence of tdTomato labeling in TRAP2;Ai14 females was used to estimate viral spread; see Materials and methods). To test this possibility, we directly compared the effects on female social behavior of caspase-mediated ablation of POA$_{social}$ neurons in TRAP2 heterozygous females (n=11 TRAP2;Ai14, a subset of those plotted in *Figure 3B–D*) and TRAP2 homozygous females (n=9, including n=4 from *Figure 3B–D*). These experiments revealed that although ablation of POA$_{social}$ neurons in TRAP2 heterozygous females tended to reduce mounting, this effect was non-significant (*Figure 3—figure supplement 1B*; McNemar's test for paired proportions; p=0.11). As in the original dataset, ablation of POA$_{social}$ neurons in TRAP2 homozygous females significantly reduced mounting (*Figure 3—figure supplement 1B*; p=0.04). Ablation of POA$_{social}$ neurons failed to reduce social investigation or USV production in either genotype, and both of these behaviors were slightly and significantly elevated following 4-OHT treatment in TRAP2 heterozygous females (*Figure 3—figure supplement 1A and C*; P<0.05 for pre-4-OHT vs. post-4-OHT social investigation and USVs in TRAP2;Ai14 females).

To ensure that ablation of POA$_{social}$ neurons in TRAP2 homozygous females was effective in eliminating POA neurons that would normally upregulate Fos following same-sex interactions in single-housed females, we compared counts of Fos-positive POA neurons in a subset of TRAP2 homozygous POA$_{social}$-caspase females to those recorded in control group-housed and single-housed females that either did or did not engage in same-sex interactions (*Figure 3—figure supplement 1D*; control female Fos data are the same as those plotted in *Figure 1F*). This analysis revealed that ablation of POA$_{social}$ neurons reduced Fos expression in the POA below the levels normally seen in either group-housed or single-housed females following same-sex interactions, down to levels observed in group-housed and single-housed females that were not given social interactions (*Figure 3—figure supplement 1D*; p<0.05 for POA$_{social}$-caspase vs. group-housed and single-housed social interaction control groups; p>0.05 for POA$_{social}$-caspase vs. group-housed and single-housed baseline control groups). Moreover, the remaining POA Fos expression in TRAP2 homozygous POA$_{social}$-caspase females was not significantly related to either rates of social investigation or to USV production (*Figure 3—figure supplement 1E, F*; linear regression; p>0.05 for both; compare to relationships between POA Fos and behavior in *Figure 1—figure supplement 1*). Taken together, these findings show that ablation of POA$_{social}$ neurons in TRAP2 homozygous females is more effective at attenuating single-housed female social behaviors than ablation in TRAP2 heterozygotes. However, in spite of a robust reduction in social behavior-driven POA Fos expression, the effects of irreversible ablation of POA$_{social}$ neurons on female social behavior differ from those of chemogenetic inhibition.

## Optogenetic activation of POA$_{social}$ neurons elicits USV production and promotes social investigation

To understand whether artificial activation of POA$_{social}$ neurons can recapitulate the effects of short-term isolation on female social behavior, we assessed the effects of optogenetic activation of POA$_{social}$ neurons on the social behaviors of group-housed females. The TRAP2 strategy was used to express

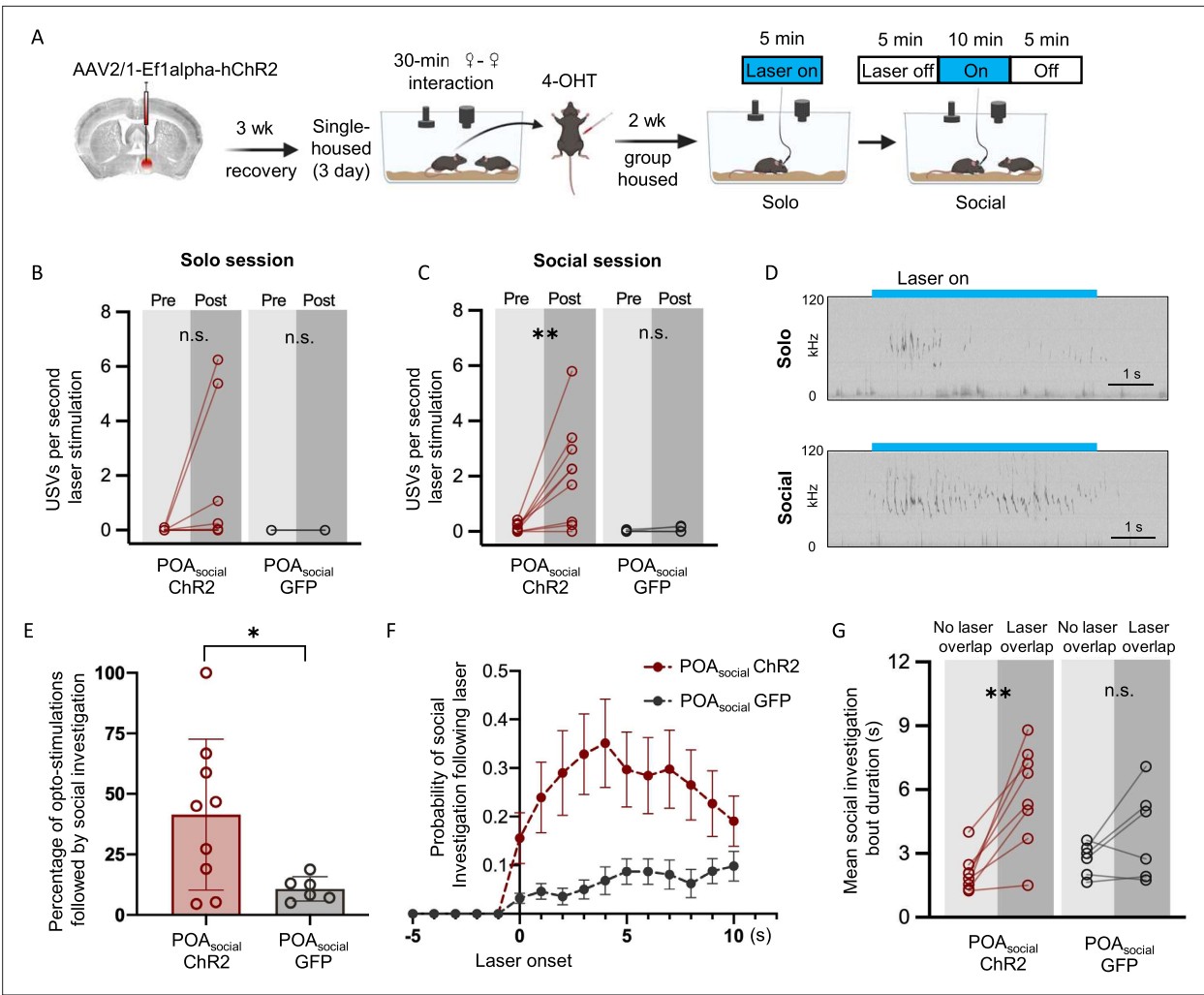

**Figure 4.** Effects of optogenetic activation of POA_social neurons on the behaviors of group-housed female mice. (**A**) Experimental timeline and viral strategy to optogenetically activate POA_social neurons in group-housed females. (**B**) Mean total USVs produced during solo sessions shown for pre-laser periods and during laser stimulation periods for experimental females with ChR2 expressed in POA_social neurons (red symbols; n=9) and for control females with GFP expressed in POA_social neurons (black symbols, n=6). (**C**) Same as (**B**), for social sessions. (**D**) Spectrograms are shown from a representative POA_social-ChR2 female to illustrate USVs that were elicited through optogenetic activation of POA_social neurons in a solo session (top) and a social session (bottom). Blue bars indicate timing of laser stimulation. (**E**) Percentage of laser stimulations followed by a bout of social investigation, plotted for POA_social-ChR2 and POA_social-GFP females. Error bars show standard deviation. (**F**) Probability of social investigation aligned with onset of laser stimulation is plotted over time for POA_social-ChR2 and POA_social-GFP females. Error bars show standard error. (**G**) Mean duration of social investigation bouts that overlapped with laser stimulation vs. bouts that did not overlap with laser stimulation is shown for POA_social-ChR2 and POA_social-GFP females.

either channelrhodopsin (ChR2) or GFP in POA_social neurons (**Figure 4A**; see Materials and methods), and females were re-group-housed for two weeks before beginning optogenetic activation experiments. The effects of optogenetically activating POA_social neurons were first assessed for each subject female in a 5-min solo session, in which the female was tested alone in a behavior chamber while pulses of blue light were delivered unilaterally to the POA (473 nm, 10 mW, 20–50 Hz, 10–20ms pulses, 5–10 s train durations). The effects of optogenetically activating POA_social neurons were then assessed for each subject female in a 20-min social session, in which a novel, group-housed female visitor was added to the behavior chamber. The pair was allowed to interact in the absence of opto-genetic stimulation for the first and last 5 min of the social session, and pulses of blue light were delivered to the POA of the subject female throughout the middle 10 min of the session (**Figure 4A**).

When POA_social-ChR2 females were tested alone, we found that optogenetic activation of POA_social neurons elicited weak-to-moderate USV production in 4 of 9 females, but the comparison of USV rates from pre-laser baseline to the laser stimulation period was not significant at the level of the entire

group (*Figure 4B*; Mann Whitney U test performed on the difference in USV rates (laser - pre-laser), p=0.09). In POA$_{social}$-GFP control females, laser stimulation failed to elicit USV production (0±0 USVs elicited in n=6 POA$_{social}$-GFP controls). Interestingly, we found that when laser stimulation was applied during social sessions, optogenetic activation of POA$_{social}$ neurons more readily elicited USV production than in solo sessions (*Figure 4C*; USVs elicited by blue laser stimulation in 8 of 9 POA$_{social}$-ChR2 females; Mann Whitney U test performed on the difference in USV rates (laser - pre-laser), p=0.006). Moreover, optogenetic activation elicited higher rates of USVs when applied at times when subject females were in close proximity to visitor females (within 2 mouse body lengths) as compared to times when the females were farther apart (mean increase in USV rates from pre-laser to laser period was 2.96±2.32 USVs/s for 'near' stimulations, 1.84±1.75 USVs/s for 'far' stimulations; paired t-test performed on the difference in USV rates (laser - pre-laser) for 'far' vs. 'near' stimulations; p=0.02). In summary, optogenetic activation of POA$_{social}$ neurons elicits USV production in group-housed females, and the efficacy of this effect is modulated by social context and proximity to a social partner.

We next considered whether optogenetic activation of POA$_{social}$ neurons modulates social investigation. We found that there was a non-significant trend toward increased overall resident-initiated investigation during the middle 10 min (laser-on period) of the social session for POA$_{social}$-ChR2 females vs. POA$_{social}$-GFP females (mean total investigation time = 72.1 ± 54.3 s for n=9 POA$_{social}$-ChR2 females vs. 23.3±16.3 s for n=6 POA$_{social}$-ChR2 females; t-test, p=0.07). To ask whether optogenetic activation could elicit social investigation, we first excluded laser stimulations in which resident-initiated social investigation occurred at any time in a 5 s window prior to laser onset. With these stimulations excluded, we then calculated the proportion of stimulations in which resident-initiated social investigation occurred following laser onset but prior to laser offset. This analysis revealed that POA$_{social}$-ChR2 females were more likely to engage in social investigation of the visitor female following laser stimulation than were POA$_{social}$-GFP females (*Figure 4E and F*; p=0.03 for difference in proportion of laser stimulations followed by social investigation). To ask whether optogenetic activation of POA$_{social}$ neurons could prolong social investigation bouts, we performed a second analysis in which social investigation bouts that occurred during the middle 10 min of the social session were separated into those that temporally overlapped with laser stimulation and those that did not. This analysis revealed that social investigation bouts that overlapped with laser stimulation were significantly longer than those without overlap in POA$_{social}$-ChR2 females but not in POA$_{social}$-GFP females (*Figure 4G*; two-way ANOVA with repeated measures on one factor; p=0.001 for overlapping vs. non-overlapping for POA$_{social}$-ChR2 females; p>0.05 for overlapping vs. non-overlapping for POA$_{social}$-GFP females).

In contrast to the effects on USV production and social investigation, optogenetic activation of POA$_{social}$ neurons only infrequently elicited mounting (activation elicited 1 bout of mounting in n=1 POA$_{social}$-ChR2 female and 0 bouts of mounting in the remaining n=8 POA$_{social}$-ChR2 females). In summary, optogenetic activation of POA$_{social}$ neurons partially mimics the effects of short-term social isolation on female behavior by promoting USV production and social investigation during same-sex interactions.

## Experiments to test whether POA neurons regulate social behaviors during same-sex interactions in group-housed females

Given our findings that POA$_{social}$ neurons promote social behaviors in single-housed females during same-sex interactions, we wondered whether a similar population of POA neurons regulates the social behaviors of group-housed females during same-sex interactions. This idea is supported by our earlier finding that Fos expression in the POA of group-housed females is significantly related to rates of resident-initiated social investigation (*Figure 1—figure supplement 1*; p=0.01) and tends to be related to rates of USV production (*Figure 1—figure supplement 1*; p=0.05) during same-sex interactions including two group-housed females.

To further test this idea, we performed two experiments to manipulate the activity of POA neurons that were TRAPed following same-sex social interactions in group-housed females (*Figure 5A and E*). In the first set of experiments, we used the TRAP2 strategy to express hM4Di in POA neurons that upregulated Fos expression following same-sex interactions in group-housed females. Two weeks following 4-OHT treatment, these females were single-housed, and the effect of CNO treatment on female social behavior during same-sex interactions was assessed. We found that chemogenetic inhibition of group-housed-TRAPed POA neurons failed to alter resident-initiated social investigation

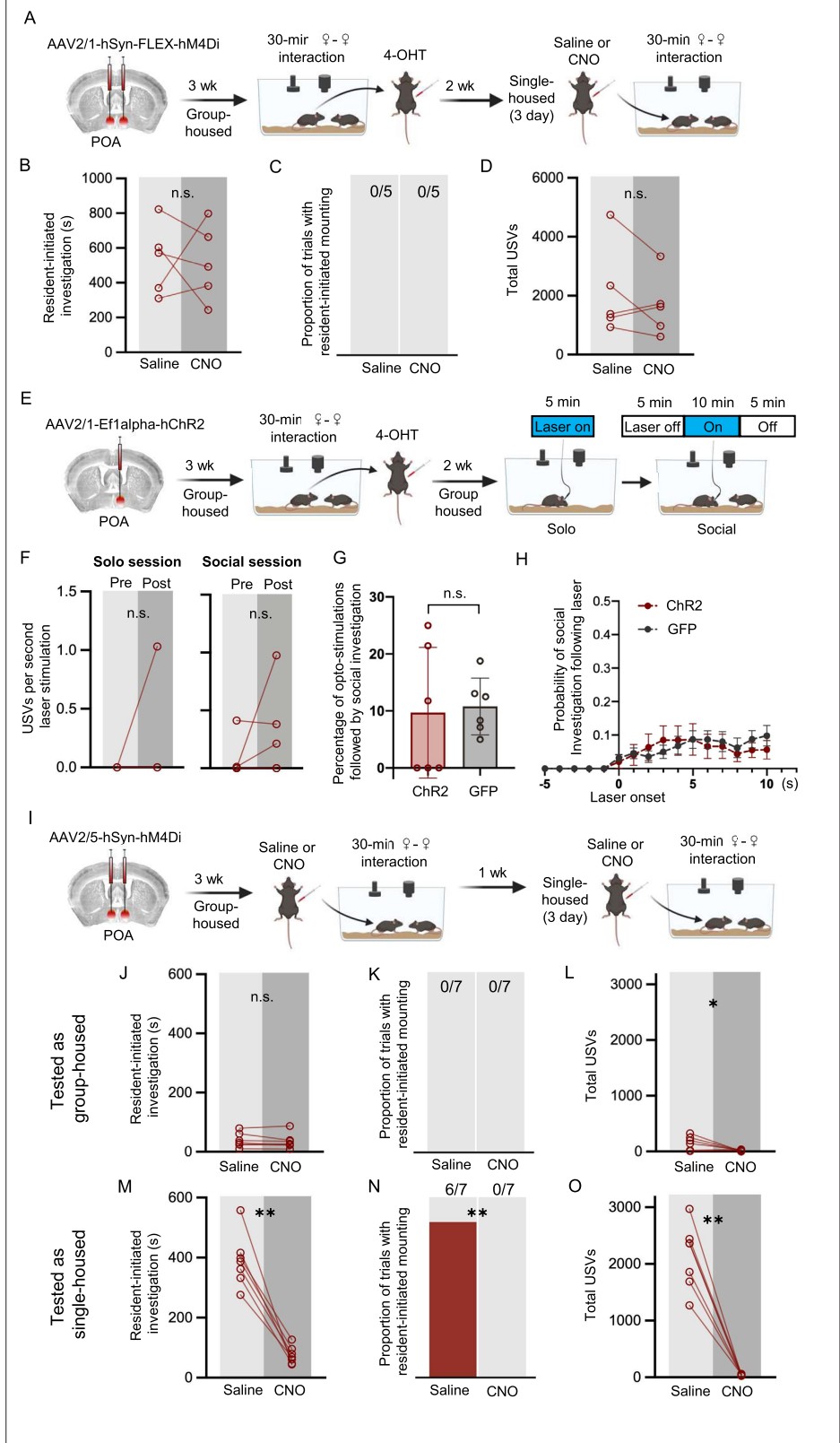

**Figure 5.** Experiments to test whether POA neurons regulate social behaviors of group-housed females.
(**A**) Experimental timeline and viral strategy to chemogenetically inhibit POA neurons that were TRAPed
following same-sex social interactions in group-housed females. (**B**) Total time spent in resident-initiated social
investigation is shown on saline and CNO days. (**C**) Same as (**B**), for proportion of trials with resident-initiated

*Figure 5 continued on next page*

*Figure 5 continued*

mounting. (**D**) Same as (**B**), for total USVs. (**E**) Experimental timeline and viral strategy to optogenetically activate POA neurons that were TRAPed following same-sex social interactions in group-housed females. (**F**) Mean total USVs produced during solo and social sessions shown for pre-laser periods and during laser stimulation periods. (**G**) Percentage of laser stimulations followed by a bout of social investigation, plotted for GH-TRAPed POA ChR2 females (n=6) and POA_social-GFP females (n=6; same control group as in *Figure 4*). Error bars show standard deviation. (**H**) Probability of social investigation following onset of laser stimulation is plotted over time for GH-TRAPed POA ChR2 and POA_social-GFP females. Error bars show standard error. (**I**) Experimental timeline and viral strategy to chemogenetically silence POA neurons in female mice, first while group-housed and a second time following single-housing. (**J**) Total time spent in resident-initiated social investigation is shown on saline and CNO days for group-housed tests. (**K**) Same as (**J**), for proportion of trials with resident-initiated mounting. (**L**) Same as (**J**), for total USVs. (**M**) Total time spent in resident-initiated social investigation is shown on saline and CNO days for single-housed tests. (**N**) Same as (**M**), for proportion of trials with resident-initiated mounting. (**O**) Same as (**M**), for total USVs.

or total USVs when females were subsequently single-housed and tested with female social partners (*Figure 5B and D*; n=5 group-housed-TRAPed hM4Di females; p>0.05 for CNO vs. saline sessions for both behaviors). Unfortunately, none of the females in this experimental cohort mounted in the saline or CNO sessions, so we were not able to evaluate the effect on mounting (*Figure 5C*). In the second set of experiments, we used the TRAP2 strategy to express ChR2 in POA neurons that upregulate Fos expression following same-sex interactions in group-housed females, and female subjects remained group-housed for subsequent tests. Optogenetic activation of group-housed-TRAPed POA neurons failed to elicit statistically significant USV production (*Figure 5F*), did not promote social investigation (*Figure 5G and H*), and did not elicit mounting (mounting observed in 0 of 6 trials).

Because group-housed females spend a relatively small amount of time engaged in social investigation, do not exhibit mounting, and produce low rates of USVs relative to single-housed females (*Figure 1B–D*), one important caveat to these experiments is that TRAP2-based activity-dependent labeling may not work efficaciously in mice that produce low rates of social behaviors. That is to say that although POA neurons may regulate social behaviors in group-housed females, these POA neurons may not strongly upregulate Fos following group-housed social interactions (see *Figure 1F*). To test whether the POA regulates the social behaviors of group-housed females using a non-activity-dependent viral strategy, we injected a non-Cre-dependent inhibitory DREAADs virus (AAV-hSyn-hM4Di) into the POA of group-housed B6 female mice (*Figure 5I*). Three weeks later, the effects of CNO treatment on social behavior during same-sex interactions were tested. We found no effects of CNO treatment on resident-initiated social investigation in group-housed females (*Figure 5J*; p>0.05), and as expected, none of the group-housed females exhibited mounting in either saline or CNO sessions (*Figure 5K*). Notably, we found that although rates of USV production were quite low, chemogenetic inhibition of POA neurons significantly reduced total USVs produced during same-sex interactions between group-housed females (*Figure 5L*; paired t-test, p=0.048). Following these tests, female subjects were single-housed for 3 days and then tested a second time. Consistent with our previous experiments, single-housing dramatically elevated rates of female social investigation, mounting, and USV production (compare saline day panels in *Figure 5J–L* vs. *Figure 5M–O*). We found that chemogenetic inactivation of POA neurons in single-housed females dramatically reduced all three behaviors (*Figure 5M–O*; paired t-tests, p<0.05 for all behaviors). These additional experiments reinforce our TRAP2-based findings that the POA regulates the social behaviors of single-housed female mice. These experiments also suggest that the POA may regulate the social behaviors of group-housed females, but that any effects may be more challenging to detect given the low rates of social behaviors produced by group-housed females during same-sex interactions.

## Characterization of female POA_social neuron molecular markers and axonal projections

Previous studies have found that USV production can be elicited in female and male mice by artificial activation of VGAT[+] POA neurons (*Gao et al., 2019*), Esr1[+] POA neurons (which are predominantly VGAT[+]; *Chen et al., 2021*; *Michael et al., 2020*), as well as POA neurons that send axonal projections to the caudolateral PAG (which are predominantly VGAT[+]; *Chen et al., 2021*; *Michael et al., 2020*). To ask to what extent POA_social neurons overlap with these previously described populations,

we first evaluated the neurotransmitter phenotype of POA$_{social}$ neurons by performing two-color in situ hybridization for c-fos mRNA and vesicular GABA transporter (VGAT) mRNA and calculating the percentage of Fos$^+$ POA neurons that co-expressed VGAT. This analysis revealed that a majority of POA$_{social}$ neurons are GABAergic (*Figure 6A and B*; n=4, 76 ± 8.8%). We next used the TRAP2 activity-dependent labeling strategy to express GFP in POA$_{social}$ neurons and found GFP-positive axons within the caudolateral PAG, indicating that at least some POA$_{social}$ neurons send axonal projections to the PAG (*Figure 6C and D*; see Materials and methods). Finally, we combined retrograde tracing from the caudolateral PAG with Fos immunostaining to quantify the percentage of PAG-projecting POA neurons that increase Fos expression in single-housed females following same-sex interactions. This experiment revealed that around 20% of PAG-projecting POA neurons express Fos in single-housed females following same-sex interactions (*Figure 6E–G*; n=4 females, percentage of tdTomato neurons that are Fos-positive=18.3 ± 2.9%). These findings suggest that a subset of POA$_{social}$ neurons overlap with GABAergic, PAG-projecting POA neurons that have been demonstrated in previous work to promote USVs via disinhibition of excitatory PAG neurons important to USV production, although only ~20% of PAG-projecting POA neurons upregulate Fos in association with female USV production (*Chen et al., 2021*; *Michael et al., 2020*).

## POA neurons increase their activity in single-housed male mice following opposite-sex but not same-sex social interactions

Given our findings that POA$_{social}$ neurons contribute to isolation-induced changes in the social behaviors of female mice, we next wondered whether a similar population of POA neurons contributes to isolation-induced changes in social behavior in male mice. To address this question, we measured the vocal and non-vocal social behaviors of sexually naive males, which were either group-housed with same-sex siblings or single-housed for 3 days and then given a 30-min social interaction with a novel, group-housed visitor. To consider the effects of isolation on male social behavior in different social contexts, males were given either a social encounter with a same-sex visitor (MM context) or with an opposite-sex visitor (MF context). Following these social interaction tests, we collected the brains of the subject males and performed immunostaining to measure Fos expression within the POA. With respect to resident-initiated investigation, we found significant main effects of both housing and social context: single-housed males spent more time investigating visitors than group-housed males, and males spent more time investigating female visitors than male visitors (*Figure 7A*; two-way ANOVA, p=0.02 for main effect of housing; p<0.001 for main effect of social context; p>0.05 for interaction effect). With respect to mounting, we found that single-housed males were more likely than group-housed males to mount female visitors, but the proportion of trials with mounting did not differ for single-housed males vs. group-housed males during male-male interactions (*Figure 7B*; z-tests for independent proportions; p=0.046 for group-housed vs. single-housed with female visitor, p>0.05 for group-housed vs. single-housed with male visitor). Similarly, there was a context-dependent effect of social isolation on male USV production, whereby only single-housed males that interacted with female visitors exhibited increased USV production relative to group-housed males (*Figure 7C*; two-way ANOVA with post-hoc Tukey's HSD tests; p<0.001 for total USVs in single-housed MF vs. group-housed MF trials; p>0.05 for total USVs in single-housed MM vs. group-housed MM trials). The finding that short-term isolation exerts larger effects on male social behavior during subsequent opposite-sex interactions relative to same-sex interactions is consistent with prior work (*Zhao et al., 2021*). When we examined POA Fos expression in these four groups of males, we found that POA Fos was significantly elevated in single-housed males following interactions with females relative to the other three groups (*Figure 7D*; two-way ANOVA, Tukey's post-hoc HSD tests; p<0.05 for difference in POA Fos between single-housed MF and all other groups). In summary, the effects of short-term isolation on male social behavior are social context-dependent, and increased Fos expression within the POA is seen in single-housed males following interactions with females, a social context marked by increased male social investigation, mounting, and USV production.

To test whether neural activity in male POA$_{social}$ neurons contributes to isolation-induced changes in male social behavior, we used the TRAP2 strategy to chemogenetically silence POA$_{social}$ neurons in single-housed males during social interactions with novel, group-housed females (see Materials and methods). The vocal and non-vocal behaviors of single-housed subject males were measured and compared during 30-min social interactions following I.P. injection of either saline or CNO. Control

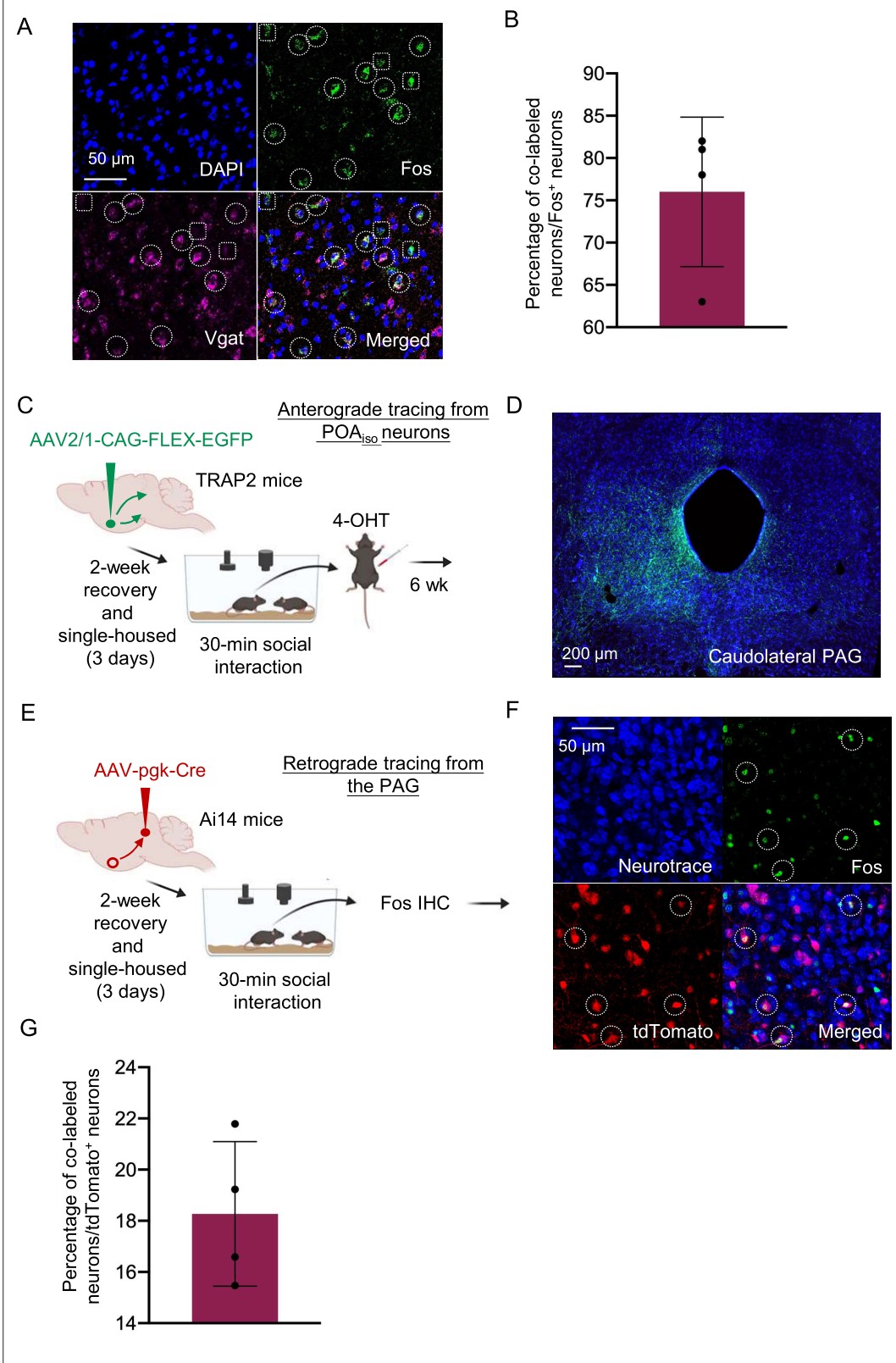

**Figure 6.** Characterization of neurotransmitter phenotype and axonal projections of POA$_{social}$ neurons. (**A**) Representative confocal images of in situ hybridization performed on brain sections containing the POA, showing overlap of expression of Fos (green) and VGAT (magenta). Blue, DAPI. (**B**) Quantification of proportion of Fos-positive POA neurons that expressed VGAT. (**C**) Experimental timeline and viral strategy to express GFP

*Figure 6 continued on next page*

in POA~social~ neurons. (**D**) Confocal images showing GFP-labeled axons of POA~social~ neurons within the caudolateral PAG. Blue, Neurotrace. (**E**) Experimental timeline and viral strategy to retrogradely label PAG-projecting POA neurons with tdTomato. (**F**) Confocal image showing tdTomato labeling in a coronal section containing the POA, and dotted circles in insets indicate examples of double-labeled neurons. Neurotrace, blue. (**G**) Quantification of proportion of tdTomato-expressing POA neurons that are also Fos-positive. All error bars show standard deviation.

males were treated identically but were injected with a virus to drive expression of GFP in POA~social~ neurons. As in females, chemogenetic inhibition of male POA~social~ neurons significantly reduced the proportion of trials with mounting (***Figure 7G***; z-test for difference between proportions; p=0.04 for proportion of trials with mounting on CNO vs. saline days in POA~social~-hM4Di males; p>0.05 for proportion of trials with mounting on CNO vs. saline days in POA~social~-GFP males). In contrast to our findings in females, chemogenetic inhibition of male POA~social~ neurons did not alter rates of resident-initiated social investigation (***Figure 7F***; two-way ANOVA with repeated measures on one factor; p<0.001 for main effect of group; p>0.05 for main effect of drug and for interaction effect) and also did not affect total USVs (***Figure 7H***; two-way ANOVA with repeated measures on one factor; p>0.05 for main effects and interaction effect). Although we are uncertain of the reason for the unexpected

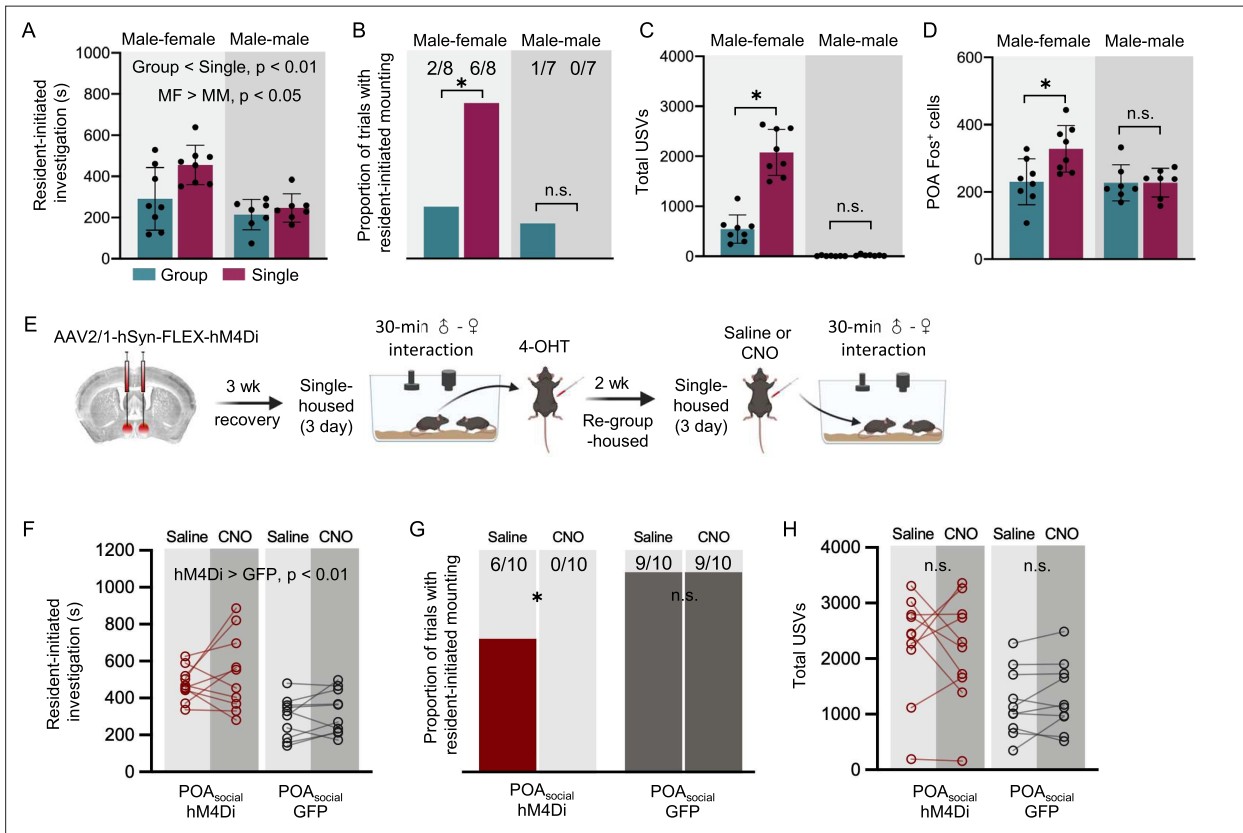

**Figure 7.** Social context-dependent differences in POA Fos expression and effects of chemogenetic inhibition of POA~social~ neurons on the social behaviors of single-housed male mice. (**A**) Total time spent in resident-initiated social investigation is shown for group-housed male residents (teal) and single-housed male residents (maroon) during interactions with either female visitors (left) or male visitors (right). (**B**) Same as (**A**), for proportion of trials with resident-initiated mounting. (**C**) Same as (**A**), for total USVs recorded from pairs containing group-housed or single-housed male residents. (**D**) Total number of Fos-positive POA neurons is shown for group-housed male residents (teal) and single-housed male residents (maroon) following interactions with female visitors (left) or male visitors (right). (**E**) Experimental timeline and viral strategy to chemogenetically inhibit the activity of POA~social~ neurons in single-housed males. (**F**) Total time spent in resident-initiated social investigation is shown on saline and CNO days for experimental males in which hM4Di is expressed in POA~social~ neurons (red symbols, n=10) and control males in which GFP is expressed in POA~social~ neurons (black symbols, n=10). (**G**) Same as (**F**), for proportion of trials with resident-initiated mounting. (**H**) Same as (**F**), for total USVs. All error bars show standard deviation.

The online version of this article includes the following figure supplement(s) for figure 7:

**Figure supplement 1.** Comparison of social behavior during TRAPing sessions for male vs. female POA~social~-hM4Di mice.

difference in social investigation times between POA$_{social}$-hM4Di and POA$_{social}$-GFP males, we note that these cohorts of mice were tested at different times and thus interacted with different groups of female visitors, which may have in turn contributed to different rates of social investigation between experimental and control males.

Given previous work implicating the POA in the regulation of male mounting (*Floody, 1989*; *Karigo et al., 2021*; *Wei et al., 2018*) as well as male USV production (*Chen et al., 2021*; *Gao et al., 2019*; *Green et al., 2018*; *Karigo et al., 2021*; *Michael et al., 2020*), we were surprised that chemogenetic silencing of POA$_{social}$ neurons in males attenuated mounting but not USV production. To begin to understand this result, we considered rates of social behaviors produced by POA$_{social}$-hM4Di males during the TRAPing sessions that preceded 4-OHT treatment and compared these to the TRAPing session behaviors produced by POA$_{social}$-hM4Di females. Interestingly, we found that POA$_{social}$-hM4Di males spent significantly more time mounting female visitors during TRAPing sessions than POA$_{social}$-hM4Di females (*Figure 7—figure supplement 1*; Mann Whitney U test; p=0.004). In comparison to females, POA$_{social}$-hM4Di males spent less time engaged in social investigation of the female visitor during TRAPing sessions (*Figure 7—figure supplement 1*; t-test; p=0.02) and also tended to produce lower rates of USVs (*Figure 7—figure supplement 1*; t-test, p=0.054). Based on these sex differences in TRAPing session social behavior, one possibility is that the bias of male TRAPing session behavior toward mounting also resulted in a bias toward TRAP2-based labeling (and subsequent chemogenetic inhibition) of POA mounting-related neurons. We note also that because all animals in the study received the same total amount of 4-OHT (150 μL of 10 mg/mL 4-OHT) and because males typically weigh more than females, the effectively smaller dosage of 4-OHT delivered to males could have reduced TRAPing efficacy relative to females and could have contributed to sex differences in the observed effects on behavior of chemogenetically silencing POA$_{social}$ neurons. In summary, we find that chemogenetic silencing of male POA$_{social}$ neurons reduces mounting during subsequent social interactions with females but does not reduce social investigation or USV production, a pattern of results that differs from the effects on single-housed female social behavior of chemogenetically silencing female POA$_{social}$ neurons.

## Discussion

In the current study, we identify and characterize a population of preoptic hypothalamic neurons that contribute to the effects of short-term social isolation on the social behaviors of mice. These POA$_{social}$ neurons exhibit increased Fos expression in single-housed female mice following same-sex social interactions, and this increase in Fos expression scales positively with the time females spend investigating and mounting female visitors and tends also to scale with rates of USVs. Chemogenetic silencing of POA$_{social}$ neurons attenuates the effects of social isolation on female social behavior, significantly reducing social investigation, mounting, and USV production. In contrast, irreversible ablation of POA$_{social}$ neurons significantly reduces mounting in single-housed females but does not reduce rates of social investigation or USVs. Optogenetic activation of POA$_{social}$ neurons partially recapitulates the effects of short-term isolation on female behavior and promotes social investigation and USV production in female mice. Finally, we extended our experiments to male mice to understand whether similar POA neurons may mediate changes in male social behavior following short-term isolation. We find that short-term isolation exerts more robust effects on male behavior during subsequent interactions with females than during subsequent interactions with males, and increased POA Fos expression is seen in single-housed males following social interactions with females but not following interactions with males. Interestingly, chemogenetic silencing of these POA$_{social}$ neurons reduces mounting but has no effect on social investigation and USV production, in contrast to the effects of chemogenetically silencing POA$_{social}$ neurons in females. Together, these experiments identify a population of preoptic hypothalamic neurons that promote social behaviors in single-housed mice in a manner that depends on sex and social context.

An extensive body of past work has implicated the POA in the regulation of male sexual behavior, including in the regulation of male courtship vocalizations in both rodents and birds (*Alger and Riters, 2006*; *Bean et al., 1981*; *Gao et al., 2019*; *Merari and Ginton, 1975*; *Riters, 2012*; *Riters et al., 2000*; *Riters and Ball, 1999*; *Wei et al., 2018*). Past work has also shown that activation of genetically-defined subsets of POA can elicit the production of USVs in both male and female mice (*Chen et al., 2021*; *Gao et al., 2019*; *Karigo et al., 2021*; *Michael et al., 2020*). However, whether

the POA regulates natural USV production in female mice remained unclear. In the current study, we demonstrate that reversible silencing of POA neurons that increase their activity during same-sex interactions decreases female USV production, indicating that the POA regulates the production of USVs by single-housed females engaged in same-sex interactions. Using a non-activity-dependent viral strategy, we also provide evidence that POA neurons regulate USV production in group-housed females. Whether these same neurons regulate female USV production in other behavioral contexts, including during interactions with male partners, remains an important open question. In addition to attenuating USV production, we found that silencing of POA_social neurons in single-housed females significantly reduced social investigation and mounting of same-sex partners. Our retrograde and anterograde tracing experiments demonstrate that at least a subset of POA_social neurons project to the caudolateral PAG where neurons important for USV production reside, and an attractive possibility is that POA_social neurons promote USV production via disinhibition of PAG-USV neurons as previously demonstrated for genetically-defined subsets of POA neurons (*Chen et al., 2021*; *Michael et al., 2020*; *Tschida et al., 2019*). Future experiments will be required to determine whether PAG-projecting POA_social neurons also regulate social investigation and mounting, or alternatively, whether distinct molecularly-defined or projection-defined subsets of POA_social neurons differentially regulate these different aspects of female social behavior. A recent study found that optogenetic activation of Tacr1$^+$ POA neurons elicits both mounting and USV production in male mice and that optogenetic activation of the axon terminals of these neurons within the PAG also promotes mounting, suggesting that some POA neurons may regulate the production of multiple social behaviors (*Bayless et al., 2023*). However, another study found that distinct subsets of male Esr1$^+$ POA neurons exhibit selective tuning in their neuronal activity to sniffing vs. mounting during interactions with females (*Yang et al., 2023*), although it remains to be tested whether these neuronal subsets functionally regulate distinct social behaviors during male-female (or female-female) interactions. The latter organization would be reminiscent of how projection-defined subsets of galanin-expressing POA neurons regulate different aspects of parental behavior (*Kohl et al., 2018*).

Although reversible inhibition of POA_social neurons reduced both social investigation and USV production in single-housed females, permanent caspase-mediated ablation of these neurons significantly reduced mounting but had no effects on social investigation or USV production. These differences are largely consistent with prior studies that reported effects on both mounting and USV production following reversible manipulations of POA activity and effects on mounting but not on USV production following irreversible manipulations of POA activity (*Bean et al., 1981*; *Chen et al., 2021*; *Gao et al., 2019*; *Karigo et al., 2021*; *Wei et al., 2018*). One possibility is that POA_social neurons do not directly regulate USV production and social investigation but rather that reversible silencing of these neurons causes off-target disruptions of neural activity in interconnected brain regions that in turn directly regulate USV production. Such a relationship was demonstrated for motor cortex, whereby reversible silencing of motor cortex disrupted performance of a learned forelimb reaching task in rats, while permanent lesions of motor cortex had no effect on task performance after learning (*Otchy et al., 2015*). However, our finding that optogenetic activation of POA_social neurons elicits USV production, along with past work demonstrating that POA activation elicits USV production (*Chen et al., 2021*; *Gao et al., 2019*; *Karigo et al., 2021*; *Michael et al., 2020*) supports the idea that the POA directly regulates USV production. Similarly, our finding that optogenetic activation of POA_social neurons promotes social investigation supports the interpretation that these neurons directly regulate this behavior. An alternative explanation for the apparently contradictory effects of chemogenetic silencing vs. ablation of POA_social neurons is that while the POA plays an obligatory role in mounting behavior (at least in certain contexts, see below), compensatory changes in additional, non-POA circuits that promote USV production and social investigation are sufficient to compensate for the permanent loss of POA_social neurons, leaving these behaviors unperturbed following POA_social neuronal ablation. Our finding that ablation of POA_social neurons dramatically reduces POA Fos expression following female-female interactions, and that the remaining Fos expression is uncorrelated with rates of USV production and social investigation, support the latter idea. The identification of forebrain-to-midbrain circuits that regulate USV production in both females and males remains an important future goal.

Our finding that POA_social neurons regulate female-female mounting extends recent work examining the role of hypothalamic regions in regulating male mounting behavior in different social contexts

(*Karigo et al., 2021*). Although the POA plays a well-established role in the regulation of mounting behavior by male rodents during interactions with females, interactions which are typically marked by high rates of USV production and are considered affiliative, the authors found that the VMH regulates male mounting behavior during interactions with other males, which is typically not accompanied by USV production and often precedes fighting. Taken together with the findings of the current study, we conclude that the POA regulates mounting behavior in both male and female mice that is directed toward female social partners and is accompanied by USV production. The question of whether the social behaviors exhibited by single-housed female mice during same-sex interactions are indeed affiliative and how these behaviors shape future interactions with female social partners are important questions that remain to be addressed. Although chemogenetic silencing and ablation of POA$_{social}$ neurons reduced female mounting, we were not able to reliably elicit mounting behavior through optogenetic activation of POA$_{social}$ neurons. We note that previous studies that elicited mounting in male mice via optogenetic activation of Esr1$^+$ POA neurons, Tacr1$^+$ POA neurons, or POA neurons that co-express Esr1 and VGAT used longer laser stimulation trains (15–30 s) than we employed in the current study (*Bayless et al., 2023*; *Karigo et al., 2021*; *Wei et al., 2018*). It is possible that longer periods of optogenetic activation of POA$_{social}$ neurons would be more effective in eliciting mounting in group-housed females, and another possibility is that POA$_{social}$ neurons overlap somewhat but not perfectly with the genetically defined groups of POA neurons manipulated in those prior studies. Regardless, our finding that optogenetic activation of POA$_{social}$ neurons promotes USV production and social investigation efficaciously relative to mounting indicates that effects on these behaviors are not secondary to effects on female mounting.

To understand whether a similar population of POA neurons regulates changes in social behavior following short-term isolation in males, we extended our experiments to single-housed males that subsequently interacted with either a male social partner or with a female social partner. Consistent with our prior work (*Zhao et al., 2021*), we found that short-term isolation exerts more robust effects on male behavior during subsequent interactions with females than during interactions with males. Although single-housed males exhibited increased rates of social investigation while interacting with both males and females relative to group-housed males, single-housed males that interacted with females exhibited increases in all measured social behaviors (social investigation, USV production, and mounting), a pattern of behavioral changes that is similar to what we observed in single-housed females. We also observed that a population of POA neurons increased Fos expression in single-housed males following interactions with female visitors but not following interactions with male visitors. Unexpectedly, we found that the effects of chemogenetically inhibiting POA$_{social}$ neurons in males differed from those in females. Namely, while reversible inhibition of POA$_{social}$ neurons reduced mounting during interactions between single-housed subject males and female visitors, there were no effects on rates of social investigation and USV production. These findings differ also from recent work showing that chemogenetic inhibition (*Chen et al., 2021*) or optogenetic inhibition (*Karigo et al., 2021*) of Esr1$^+$ POA neurons reduces USV production in male mice. Although the factors that account for this difference in results are unclear, one possibility is that our TRAP2-based viral labeling in males is biased toward POA neurons that regulate mounting as compared to POA neurons that regulate USV production, although as stated above, more work is required to understand whether these different social behaviors are regulated by distinct or overlapping subsets of POA neurons. Future experiments can also explore to what extent female and male POA$_{social}$ neurons represent molecularly, anatomically, and functionally similar or dissimilar neuronal populations.

The current study adds to an emerging body of literature implicating the POA in the regulation of social behavior in both females and males, including in non-sexual social contexts (*Fukumitsu et al., 2022*; *Hu et al., 2021*; *Liu et al., 2023*; *McHenry et al., 2017*; *Wu et al., 2021*). Our findings also complement recent work that has described a role for the POA in regulating behavioral responses to social isolation in female mice. A recent study using BALB/c mice found that reunion with same-sex cagemates following short-term 'somatic isolation' (i.e. separation from cagemates via a partition that permits visual, auditory, and olfactory signaling but prevents physical contact) increases the activity of calcitonin-receptor (Calcr) expressing POA neurons (*Fukumitsu et al., 2023*). Knockdown of Calcr expression in these neurons reduced social-seeking behaviors directed at the cage partition exhibited by single-housed females, and chemogenetic activation of these neurons increased partition-biting behavior in single-housed females. Another recent study using female FVB/NJ mice applied a TRAP2-based intersectional approach to identify POA neurons that increase their activity following reunion with same-sex cagemates (MPN$_{reunion}$

neurons; *Liu et al., 2023*). Optogenetic activation of these neurons did not alter social investigation in group-housed females, whereas activation of these neurons during reunion with cagemates following short-term social isolation decreased social investigation. However, optogenetic inhibition of $MPN_{reunion}$ during reunion did not alter rates of social investigation in single-housed females. Although we did not test the effects of activating $POA_{social}$ neurons in single-housed females in the current study, the difference in the effects of inhibiting $POA_{social}$ neurons on the behavior of single-housed females (reduced rates of social investigation, USV production, and mounting) relative to the effects of optogenetic inhibition of $MPN_{reunion}$ neurons in Liu et al. (no effect on social investigation during reunion) suggest that these represent distinct populations of POA neurons. We note that in the current study, single-housed female subjects remained single-housed for 24 hr following social interaction TRAPing sessions, a design intended to maximize viral labeling of POA neurons that promote increased female social behaviors and that would likely in turn minimize viral labeling of POA neurons that promote social satiety. These studies highlight a complex role for the POA in regulating multiple aspects of changes in social behavior following short-term social isolation. Although some studies of social isolation have avoided the use of C57BL/6 J female mice, the robust triad of changes in social behavior exhibited by these females following short-term isolation affords a powerful opportunity to continue investigating neural circuit mechanisms through which short-term social isolation promotes social behaviors, as well as to investigate how hypothalamic circuits regulate the coordinated production of suites of social behaviors during female-female social interactions.

# Materials and methods

## Key resources table

| Reagent type (species) or resource | Designation | Source or reference | Identifiers | Additional information |
|---|---|---|---|---|
| Transfected construct (*Mus musculus*) | *TRAP2 Fos^{tm2.1(icre/ERT2)Luo}*/J | Jackson Laboratories | RRID:IMSR_JAX:030323 | |
| Transfected construct (*M. musculus*) | Ai14 B6.Cg*Gt(ROSA)26Sor^{tm14(CAG-tdTomato)Hze}*/J | Jackson Laboratories | RRID:IMSR_JAX:007914 | |
| Transfected construct (*M. musculus*) | C57BL/6 J | Jackson Laboratories | RRID:IMSR_JAX:000664 | |
| Antibody | c-Fos (9F6) Rabbit mAb (rabbit monoclonal) | Cell Signaling Technology | RRID:AB_2247211 CAT#:2250S | 1:1000 |
| Antibody | Alexa Fluor 488 goat-anti-rabbit (goat polyclonal) | Invitrogen | RRID:AB_143165 CAT#:A-11008 | 1:1000 |
| Recombinant DNA reagent | AAV2/1-hSyn-FLEX-hM4Di-mCherry | Addgene | RRID:Addgene_44262 | |
| Recombinant DNA reagent | AAV2/1-Ef1alpha-hChR2(h134R)-EYFP-WPRE-HGHpA | Addgene | RRID:Addgene_20298 | |
| Recombinant DNA reagent | AAVrg-pgk-Cre | Addgene | RRID:Addgene_24593 | |
| Recombinant DNA reagent | pAAV2/5-flex-taCasp3-TEVp | Addgene | RRID:Addgene_45580 | |
| Recombinant DNA reagent | pAAV2/5-hSyn-hM4Di-mCherry | Addgene | RRID:Addgene_50475 | |
| Recombinant DNA reagent | AAV2/1-pCAG-FLEX-EGFP-WPRE | Addgene | RRID:Addgene_51502 | |
| Commercial assay or kit | HCR v3.0 | Molecular Instruments | | |
| Chemical compound, drug | Clozapine N-oxide dihydrochloride | Hello Bio | CAT#:HB6149 | |
| Chemical compound, drug | 4-Hydroxytamoxifen | Hello Bio | CAT#:HB6040 | |
| Software, algorithm | MATLAB | Mathworks | RRID:SCR_001622 | |
| Software, algorithm | Zen | Zeiss | RRID:SCR_013672 | |
| Software, algorithm | Spike2 | CED | RRID:SCR_000903 | http://ced.co.uk |
| Software, algorithm | ImageJ | NIH | RRID:SCR_003070 | https://imagej.net/ij/ |
| Software, algorithm | Behavioral Observation Research Interactive Software (v. 8.13) | Open Behavior | RRID:SCR_021509 | https://github.com/olivierfriard/BORIS |
| Software, algorithm | R Project for Statistical Computing | R Core Team | RRID:SCR_001905 | http://www.r-project.org/ |
| Software, algorithm | R Studio | Posit | RRID:SCR_000432 | https://posit.co/ |
| Other | Neurotrace 435/455 Blue | Thermo Fischer Scientific | CAT#:N21479 | 1:500 |

### Lead contact

Further information and requests for resources should be directed to and will be fulfilled by the lead contact, Katherine Tschida (kat227@cornell.edu).

## Experimental models and subject details

### Animal statement

All experiments and procedures were conducted according to protocols approved by the Cornell University Institutional Animal Care and Use Committee (protocol #2020–001).

### Animals

TRAP2 (Jackson Laboratories, 030323), Ai14 (Jackson Laboratories, 007914), TRAP2;Ai14, and C57Bl/6 J (Jackson Laboratories, 000664) mice were at least 8 weeks old at the time of the experiments or surgeries. TRAP2;Ai14 mice were generated by crossing TRAP2 with Ai14. All mice were kept on a 12:12 reversed light/dark cycle, were housed in ventilated micro-isolator cages in a controlled environment with regulated temperature and humidity and were provided with unrestricted access to food and water. A running wheel (Innovive) was present in all home cages from the time of weaning and was subsequently removed immediately before initiating the social interaction test. Mouse cages were cleaned weekly, and experiments were never conducted on cage change days.

TRAP2 female and male subjects were used in all experiments, with the exception of n=11 TRAP2;Ai14 females subjects used in the neuronal ablation experiments (*Figure 3*, *Figure 3—figure supplement 1*), n=4 Ai14 females used in anatomical experiments to examine overlap between PAG-projecting POA neurons and POA neurons that upregulate Fos following same-sex interactions in single-housed females (*Figure 6*), and n=7 B6 females used for non-Cre-dependent chemogenetic silencing of POA (*Figure 5I–O*).

## Methods details

### Social isolation and social interaction tests

Female and male subject mice were either group-housed with same-sex siblings or separated from their cage mates and individually housed in clean cages for 3 days prior to behavioral tests. In the case of group-housed subject mice, siblings were temporarily removed from the home cage for the duration of the test. The subject animal's home cage was then placed in a sound-attenuating recording chamber (Med Associates) equipped with an ultrasonic microphone (Avisoft), an infrared light source (Tendelux), and a webcam (Logitech, with the infrared filter removed to enable video recording under infrared lighting conditions). A novel, group-housed visitor mouse (female or male mouse on a C57BL/6 background) was placed in the home cage of the subject mouse, and vocal and non-vocal behaviors were recorded for 30 min. Visitor mice were used across multiple experiments (<6 in total), including in interactions with both group-housed and single-housed subject mice. Visitor females used in male-female interactions were never used for female-female experiments, but a subset of visitors used in female-female interactions were subsequently used in male-female experiments.

A separate cohort of female mice was used to investigate the effects on social behavior of re-group-housing following a period of social isolation. For this cohort of mice, social interaction tests with novel, group-housed female visitors were conducted at three timepoints: (1) on day 0, when subject females were still group-housed; (2) On day 3, after being single-housed for 3 days; (3) on day 17, after a randomly selected subset of subject females were re-group-housed with their siblings for 2 weeks, and the remaining female subjects remained single-housed for 2 weeks.

### USV recording and detection

USVs were recorded with an ultrasonic microphone (Avisoft, CMPA/CM16), amplified (Presonus TubePreV2), and digitized at 250 kHz (Avisoft UltrasoundGate 166 H or CED Power 1401). USVs were detected with custom MATLAB codes (*Tschida et al., 2019*) using the following parameters (mean frequency >45 kHz; spectral purity >0.3; spectral discontinuity <1.00; minimum USV duration = 5ms; minimum inter-syllable interval = 30ms).

## Analyses of non-vocal social behaviors

Trained observers used BORIS software (v.8.13) to score the following non-vocal behaviors: resident-initiated social investigation and resident-initiated mounting. Social investigation included sniffing and following. Resident-initiated mounting of the visitor typically occurred following a period of resident-initiated social investigation, with the resident mouse positioning its forelimbs on top of the body of the visitor, sometimes with pelvic thrusts and sometimes without. Neither visitor-initiated mounting nor fighting were observed in our dataset.

In some trials, total movement was estimated using a custom MATLAB code that allows the user to mark the position of a mouse in every 30th frame (i.e.,once per second). Total movement was then calculated as the sum of changes in position across pairs of marked frames.

## Fos immunohistochemistry

Two hours following the start of the social interaction test (or from the start of the solo behavior session for the no-interaction, baseline groups of mice), mice were deeply anesthetized using isoflurane and then transcardially perfused with phosphate-buffered saline (PBS, pH 7.4), followed by 4% paraformaldehyde (PFA; Sigma-Aldrich, in 0.1 M PBS, pH 7.4). Brains were subsequently dissected and post-fixed in 4% PFA for 24 hr at 4 °C, followed by immersion in 30% sucrose solution in PBS for 48 hr at 4 °C. Afterward, brains were embedded in frozen section embedding medium (Surgipath, VWR), flash frozen in a dry ice-ethanol (100%) bath, and then stored at –80 °C until sectioning. Sections were cut on a cryostat (Leica CM1950) to a thickness of 80 μm, washed in PBS (3x5 min at RT), permeabilized for 2–3 hr in PBS containing 1% Triton X-100 (PBST), and then blocked in 0.3% PBST containing 10% Blocking One (Nacalai USA) for 1 hr at RT on a shaker. Sections were then incubated for 24 hr at 4 °C with primary antibody in blocking solution (1:1000 rabbit-anti-Fos, Cell Signaling Technologies, 2250 S), washed 3x30 min in 0.3% PBST, then incubated for 24 hr at 4 °C with secondary antibody in blocking solution (1:1000, Alexa Fluor 488 goat-anti-rabbit, Invitrogen, plus 1:500 NeuroTrace, Invitrogen) Finally, sections were washed for 2x10 min in 0.3% PBST, followed by washing for 2x10 min in PBS. After mounting on slides, sections were dried and coverslipped with Fluromount G (Southern Biotech). Slides were imaged with a 10 x objective on a Zeiss 900 laser scanning confocal microscope, and Fos-positive neurons within regions of interest were counted manually by trained observers.

## Floating section two-color in situ hybridization

In situ hybridization was conducted using hybridization chain reaction (HCR v3.0, Molecular Instruments). Ten minutes after the completion of the 30-min social interaction tests, mice underwent transcardial perfusion with RNase-free PBS (DEPC-treated), followed by 4% PFA. Dissected brain samples were post-fixed overnight in 4% PFA at 4 °C, cryoprotected in a 30% sucrose solution in DEPC-PBS at 4 °C for 48 hr, flash frozen in section embedding medium, and stored at –80 °C until sectioning. 40-μm-thick coronal floating sections were collected into sterile 24-well plates in DEPC-PBS. These sections were briefly fixed once again for 5 min in 4% PFA and subsequently immersed in 70% EtOH in DEPC-PBS overnight. Sections were then rinsed in DEPC-PBS, incubated for 45 min in 5% SDS in DEPC-PBS, followed by a series of rinses and incubations: 2 x SSCT, pre-incubation in HCR hybridization buffer at 37 °C, and incubation in HCR hybridization buffer containing RNA probes (VGAT and Fos) overnight at 37 °C. Sections were then rinsed 4x15 min at 37 °C in HCR probe wash buffer, rinsed in 2 X SSCT, pre-incubated in HCR amplification buffer, and then incubated in HCR amplification buffer containing HCR amplifiers at RT for approximately 48 hr. On the final day, sections were rinsed in 2 x SSCT, counterstained with DAPI (Thermo Fisher, 1:5000), rinsed again with 2 x SSCT, mounted on slides, and coverslipped with Fluoromount-G (Southern Biotech). After drying, slides were imaged with a 10 x or 20 x objective on a Zeiss 900 laser scanning microscope. Neurons were scored from three sections of tissue from the POA from each mouse, and the absence of presence of staining for different probes was quantified manually by trained scorers.

## Viruses

The following viruses and injection volumes were used: AAV2/1-hSyn-FLEX-hM4Di-mCherry (Addgene #44262, 200 nL), AAV2/1-CAG-FLEX-EGFP-WPRE (Addgene #51502, 200 nL), AAV2/5-Ef1alpha-FLEX-taCasp3-TEVp (Addgene #45580, 200 nL), AAV2/1-Ef1alpha-hChR2(h134R)-EYFP-WPRE-HGHpA

(Addgene #20298, 200 nL), AAVrg-pgk-Cre (Addgene #24593, 200 nL), and AAV2/5-hSyn-hM4Di-mCherry (Addgene #50475, 200 nL). The final injection coordinates were as follows: POA, AP = –0.1 mm, ML = 0.6 mm, DV = 5.1 mm; AH, AP = –0.7 mm, ML = 0.6 mm, DV = 5.1 mm; VMH, AP = –1.5 mm, ML = 0.7 mm, DV = 5.4 mm; PAG, AP = –4.7 mm, ML = 0.6 mm, DV = 1.75 mm. Viruses were pressure-injected using a pulled glass pipettes mounted in a programmable nanoliter injector (Nanoject III, Drummond) at a rate of 15 nL every 60 s.

## Stereotaxic surgery

Mice were anesthetized using isoflurane (2.5% for induction, then 1.5–2.5% for maintenance) and then securely positioned in a stereotaxic apparatus (Angle Two, Leica). A midline incision in the scalp was made to expose the skull, and small craniotomies were created dorsal to each injection site using a surgical drill. Viral injection pipettes were left in place for a minimum of 10 min before and after viral injections to minimize backflow upon pipette withdrawal from the brain. Surgical sutures (LOOK 774B, Fisher Scientific) and tissue adhesive (3 M) were used to close the incision.

For optogenetic activation experiments, an optogenetic ferrule (RWD Fiber Optic Cannula, Ø1.25 mm Ceramic Ferrule, 200 μm Core, 0.22 NA, L=7 mm) was implanted approximately 250 μm above the viral injection site immediately following the viral injection and was secured in place with Metabond (Parkell).

## TRAP activity-dependent labeling

Solutions of 4-hydroxytamoxifen (4-OHT, HelloBio, HB6040) were prepared by dissolving 4-OHT powder at 20 mg/mL in ethanol by shaking at 37 °C, and aliquots (75 μL) were then stored at –20 °C. Before use, 4-OHT was redissolved in ethanol by shaking at 37 °C and filtered corn oil was added (150 μL). Ethanol was then evaporated by vacuum under centrifugation to give a final concentration of 10 mg/mL, and the 4-OHT solution was used on the same day it was prepared.

To express viral transgenes in recently active neurons, we used the Targeted Recombination in Active Populations (TRAP2) strategy. Three weeks following viral injection, TRAP2 and TRAP2;Ai14 mice were single-housed for 3 days and then given 30-min social encounters (as described above). Following the social encounter, subject mice received I.P. injections of 4-OHT (150 μL of 10 mg/mL 4-OHT in filtered corn oil) to enable expression of viral transgenes in recently active neurons. To minimize neural activity triggered by stimuli outside of the social interaction test, all subject animals were individually housed for an additional 24 hr following 4-OHT treatment before being re-group-housed with their same-sex siblings. In some control experiments, mice remained group-housed prior to the 30-min social encounter and 4-OHT treatment (*Figure 5A–H*) or were single-housed but were not given a social encounter prior to 4-OHT treatment (*Figure 2—figure supplement 1D–G*).

## Chemogenetic inhibition

To reversibly reduce neuronal activity, TRAP2 female mice received bilateral injections of an Cre-dependent inhibitory DREADDs virus into the hypothalamus (AAV2/1-hSyn-FLEX-hM4Di-mCherry; injected into the POA, AH, or VMH) as described above. TRAP2 male mice received the same viral injections into the POA only. Three weeks later, mice were single-housed for 3 days and then were subsequently given a 30 min social encounter in their home cage with a novel, group-housed female visitor. Subject mice then received an I.P. injection of 4-OHT to enable expression of hM4Di in activity-defined populations of hypothalamic neurons. Following the TRAPing session, subjects remained single-housed for an additional 24 hr. Subjects were then re-group-housed with same-sex siblings for 2 weeks and were single-housed a second time prior to behavioral testing. On the first day of testing, subject mice received an I.P. injection of either sterile saline (as a control) or clozapine-n-oxide (CNO, 4 mg/kg, Hello Bio HB6149; to inhibit neurons expressing hM4Di) 30 minutes prior to a social interaction test. Following this first test, mice remained single-housed. Three days later, mice that previously were treated with saline received an I.P. injection of CNO, and mice that were previously treated with CNO received an I.P. injection of saline, 30 min prior to another social interaction. Rates of USV production and non-vocal social behaviors were compared between saline and CNO days within animals to assess the effects of neuronal inhibition on social behaviors. Control mice received unilateral injections into the POA of a Cre-dependent AAV driving the expression of GFP (AAV2/1-CAG-FLEX-EGFP) and were otherwise treated identically to experimental animals.

## Neuronal ablation

To permanently ablate neurons, TRAP2;Ai14 or TRAP2 female mice received bilateral injections of an AAV2/5-ef1alpha-FLEX-taCasp3-TEVp virus into the POA. Following a three-week recovery period, these animals were individually housed for three days and subsequently given a 30-min social encounter in their home cage with a novel, group-housed female visitor. Subject mice then received an I.P. injection of 4-OHT to enable expression of caspase in activity-defined POA neurons. Two weeks later, females were single-housed for 3 days and then given a second 30min social interaction test. Social behaviors of subject females were compared between the pre-ablation and post-ablation interaction tests to assess the effects of neuronal ablation.

## Optogenetic activation

Female TRAP2 mice received unilateral injections into the POA of AAV-ef1α-FLEX-ChR2 (experimental) or AAV-CAG-FLEX-GFP (control). In the same surgery, an optogenetic ferrule was implanted approximately 250 μm above the viral injection site. Three weeks later, females were single-housed for 3 days and then given a 30-min social interaction with a novel group-housed, female visitor. Subject females then received an I.P. injection of 4-OHT. Two weeks later, females were first placed alone in a clean testing chamber for a 5-min habituation period after connecting the laser patch cable to the female's optogenetic ferrule. Optogenetic activation sessions consisted of a 5-min period in which optogenetic activation was performed in solo females, followed by a 20-min period in which activation was performed as subject females interacted with a novel, group-housed female visitor. The social session was further divided into three phases: 5 min without optogenetic activation, 10 min with optogenetic activation, and 5 min without optogenetic activation. During the middle 10 min of the social session, some laser stimuli were delivered at times when the two females were near to one another (inter-animal distance <~2 mouse body lengths), and other stimuli were delivered at times when the females were not in close contact. POA$_{social}$ neurons were optogenetically activated with illumination from a 473 nM laser (10 mW) at 20–50 Hz (10–20ms pulses, trains lasted 5–10 s) Laser stimuli were driven by computer-controlled voltage pulses (Spike 2 version 10.8, CED).

## Anatomical tracing

Female TRAP2 mice used as GFP controls in the chemogenetic inhibition experiments were subsequently used for anterograde mapping of the axonal projections of POA$_{social}$ neurons. Three weeks following unilateral injection into the POA of a Cre-dependent AAV driving the expression of GFP (AAV2/1-CAG-FLEX-EGFP), females were single-housed for 3 days and subsequently given a 30-min social interaction test. Subject mice then received an I.P. injection of 4-OHT. Six weeks later, females were perfused, brains were collected and sectioned, and a confocal microscope was used to image GFP-positive axon terminals within coronal brain tissue sections.

To examine the overlap between PAG-projecting POA neurons and Fos expression, Ai14 females first received a unilateral injection into the PAG of an AAV driving the retrograde expression of Cre-recombinase (AAVrg-pgk-Cre). Two weeks later, these females were given a 30-min social interaction test. Ninety minutes after the test, subject females were perfused, brains were collected, and coronal brain sections containing the POA were collected for Fos immunohistochemistry as described above. Brain tissue sections were imaged with a 10 x objective on a Zeiss 900 laser scanning confocal microscope, and POA neurons that were Fos-positive and tdTomato-positive were counted manually by trained observers.

## Statistics

Two-sided statistical comparisons were used in all analyses (alpha = 0.05). The Shapiro-Wilk test was performed to analyze the normality of each data distribution, and non-parametric statistical tests were used for comparisons that included non-normally distributed data. No statistical methods were used to pre-determine sample size. Mice were randomly assigned to either experimental or control groups. Video scoring of non-vocal social behaviors was conducted by trained scorers that were blinded to group identity. Mice were only excluded from analysis in cases in which viral injections were not targeted accurately. Details of the statistical analyses used in this study are included in *Supplementary file 1*.

## Acknowledgements

Thanks to Frank Drank and other CARE staff for their excellent mouse husbandry. All experimental design schematics included in the figures were created using icons that were generated with BioRender.com.

## Additional information

### Funding

| Funder | Grant reference number | Author |
|---|---|---|
| National Institute of Mental Health | R01MH136887 | Katherine Tschida |

The funders had no role in study design, data collection and interpretation, or the decision to submit the work for publication.

### Author contributions

Xin Zhao, Conceptualization, Data curation, Formal analysis, Investigation, Visualization, Writing – original draft, Writing – review and editing; Yurim Chae, Destiny Smith, Valerie Chen, Dylan DeFelipe, Joshua W Sokol, Archana Sadangi, Investigation; Katherine Tschida, Conceptualization, Data curation, Formal analysis, Supervision, Funding acquisition, Visualization, Methodology, Writing – original draft, Project administration, Writing – review and editing

### Author ORCIDs

Xin Zhao https://orcid.org/0000-0002-5796-8034
Katherine Tschida https://orcid.org/0000-0002-8171-1722

### Ethics

This study was performed in strict accordance with the recommendations in the Guide for the Care and Use of Laboratory Animals of the National Institutes of Health. All of the animals were handled according to approved institutional animal care and use committee (IACUC) protocols (#2020-001) of Cornell University. All surgery was performed under isoflurane anesthesia, and every effort was made to minimize suffering.

Reviewer #2 (Public review): https://doi.org/10.7554/eLife.94924.3.sa1
Reviewer #3 (Public review): https://doi.org/10.7554/eLife.94924.3.sa2
Reviewer #4 (Public review): https://doi.org/10.7554/eLife.94924.3.sa3
Author response https://doi.org/10.7554/eLife.94924.3.sa4

## Additional files

### Supplementary files

MDAR checklist
Supplementary file 1. Details of statistical analyses.

### Data availability

All data generated and analyzed during this study have been deposited at Cornell eCommons and are publicly available.

The following dataset was generated:

| Author(s) | Year | Dataset title | Dataset URL | Database and Identifier |
|---|---|---|---|---|
| Xin Z, Yurim C, Smith D, Chen V, DeFelipe D, Sokol J, Sadangi A, Tschida K | 2025 | Data from: Short-term social isolation acts on hypothalamic neurons to promote social behavior in a sex- and context-dependent manner | https://doi.org/10.7298/cgfv-zz39 | Cornell University eCommons Repository, 10.7298/cgfv-zz39 |

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
