## [Editor Report · eLife Assessment]

This **important** study substantially advances our understanding of the neural circuits that regulate social behavior by identifying a population of hypothalamic neurons in the preoptic area that promote social interactions following short-term isolation. The evidence supporting the authors' claims is **solid**, with well-designed experiments using validated activity-dependent tagging and manipulation methods, though some differences in outcomes between experiments highlight limitations of the tagging approach. The work will be of broad interest to neuroscientists studying social behavior, neural circuit function, and hypothalamic mechanisms and will represent a meaningful contribution to the field.

---

## [Referee Report · Reviewer #2 (Public review)]

Summary:

This study reveals that short-term social isolation increases social behavior at a reunion, and a population of hypothalamic preoptic area neurons become active after social interaction following short-term isolation (POAsocial neurons). Effectively utilizing a TRAP activity-dependent labeling method, the authors inhibit or activate the POAsocial neurons and find that these neurons are involved in controlling various social behaviors, including ultrasonic vocalization, investigation, and mounting in both male and female mice. This work suggests a complex role for the POA in regulating multiple aspects of social behavior, beyond solely controlling male sexual behaviors.

Strengths

While a few studies have shown that optogenetic activation of the POA in females promotes vocalization and mounting behavior similar to the effects observed in males, these were results of artificially stimulating POA neurons, and whether POA neurons play a role in naturally occurring female social behaviors was unknown. This paper clearly demonstrates that a population of POA neurons is necessary for naturally evoked female social vocalizations and mounting behaviors.

Weaknesses

The authors used various gain-of-function and loss-of-function methods to identify the function of POAsocial neurons. However, there were inconsistent results among the different methodologies. As the authors describe in the manuscript, these inconsistencies are potentially due to limitations of the TRAP activity-dependent labeling method; however, different approaches will be necessary to clarify these issues.

Overall, this paper is well-written and provides valuable new data on the neural circuit for female social behaviors and the potentially complex role of POA in social behavior control.

---

## [Referee Report · Reviewer #3 (Public review)]

Summary:

The mechanisms by which short-term isolation influences the brain to promote social behavior remain poorly understood. The authors observed that acute isolation enhanced social behaviors, including increased investigation, mounting, and ultrasonic vocalizations (USVs). These effects were evident in same-sex interactions among females and in male-female interactions. Concurrently, cFos expression in the preoptic area (POA) of the hypothalamus was selectively elevated in single-housed females. To further investigate, the authors used an innovative tagging strategy (TRAP2) to manipulate these neurons. Overall, the study identifies a population of hypothalamic neurons that promote various aspects of social behavior after short-term isolation, with effects that are sex- and context-dependent.

Strengths:

Understanding the neural circuit mechanisms underlying acute social isolation is an important and timely topic. By employing state-of-the-art techniques to tag neurons active during specific behavioral epochs, the authors identified the preoptic area (POA) as a key locus mediating the effects of social isolation. The experimental design is sound, and the data are of high quality. Notably, the control experiments, which show that chemogenetic inactivation of other hypothalamic regions (AH and VMH) does not affect social behavior, strongly support the specificity of the POA's role within the hypothalamus. Through a combination of behavioral assays, activity-dependent neural tagging, and circuit manipulation techniques, the authors provide compelling evidence for the POA's involvement in behaviors following social isolation. These findings represent a valuable contribution to understanding how hypothalamic circuits adapt to the challenges of social isolation.

Weaknesses:

The authors conducted several circuit perturbation experiments, including chemogenetics, ablation, and optogenetics, to investigate the effects of POA-social neurons. They observed that the outcomes of these manipulations varied depending on whether the intervention was chronic (e.g., ablations) or acute (e.g., DREADDs), potentially due to compensatory mechanisms in other brain regions. Furthermore, their additional experiments revealed that the robustness of the manipulations was influenced by the heterozygosity or homozygosity of TRAP2 animals. While these findings suggest that POA neurons contribute to multiple behavioral responses to social isolation, further experiments are needed to clarify their precise roles.

---

## [Referee Report · Reviewer #4 (Public review)]

Summary:

Using immunostaining for the immediate early gene Fos, and employing TRAP2-mediated chemogenetic and optogenetic perturbations, the authors provide evidence that neurons in the preoptic hypothalamus, identified as 'POA-social neurons,' promote social behaviors in mice - particularly in socially isolated (or deprived) mice, who exhibit an increased motivation for social investigations.

Strengths:

The focus on female-female social interactions is a valuable contribution to the field, as these interactions are less studied and the underlying neural mechanisms are less understood. The authors should be commended for their comprehensive approach in performing and reporting multiple perturbation experiments, including optogenetics, chemogenetics, and ablation. The authors also deserve recognition for their thoughtful discussion of the nuances in the phenotypes observed across these various perturbation experiments.

Weaknesses:

A limitation of the paper, however, is the insufficient clarification of the specific functions of these POA-social neurons. In my interpretation of the results, the neurons may be crucial for motivated social behaviors in females and motivated mounting of females in males, regardless of whether the test mice are housed singly or in groups. For group-housed mice, the motivation to interact with stimulus mice was likely low in their behavioral paradigm, which may explain the reduced interactions observed in the resident-intruder assay and why these neurons were not tagged (TRAPed) in that setting. Tagging these neurons in singly housed mice following a social interaction, followed by imaging in a group setting where motivated social behaviors do occur, could elucidate whether these neurons are specifically activated during social interactions in socially deprived mice or are generally crucial for motivated social behaviors in any setting. I understand that such calcium imaging may be beyond the scope of this version of the paper, but incorporating these results in a future version would significantly enhance the paper's impact. Depending on the outcomes of such experiments, the title 'Short-term social isolation acts on hypothalamic neurons to promote social behaviors in a sex- and context-dependent manner' may need to be revised to more accurately reflect the findings.

---

## [Author Response]

The following is the authors’ response to the original reviews.

**Reviewer 1:**
The main weaknesses of the paper are a lack of significance in key findings, and relatedly, concluding effects from insignificant findings. Additional elements could be improved to help strengthen this overall well-rounded and intriguing set of results.

In the original manuscript, we reported that chemogenetic silencing of POA-social neurons (previously called POA-iso neurons; more details on rationale for renaming below in our responses to reviewer recommendations) tended to reduce mounting in both single-housed female and single-housed male mice, although these effects were non-significant. We have added samples to both datasets and now report that chemogenetic silencing of POA-social neurons significantly reduces the proportion of trials with mounting in both sexes (Fig. 2C and Fig. 6G).

We have also included new analyses to test whether optogenetic activation of POAsocial neurons in group-housed females promotes social investigation (in addition to USV production, as reported in the original manuscript). We now report that optogenetic activation of POA-social neurons significantly increases the probability of social investigation (Fig. 4E-F) and significantly increases the duration of social investigation bouts (Fig. 4G).

Additional recommendations from the reviewer are addressed in detail below. Thank you for your critical and insightful feedback.

**Reviewer 2:**
All the activity-dependent labeling experiments with TRAP mice, including the subsequent neural activity manipulation experiments (Figures 2, 3, 4, 5E-F), were conducted by labeling neurons only in socially isolated animals, not group-housed animals. The authors labeled neurons after 30-minute social interactions, raising the possibility that the labeled neurons simply represent a "social interaction/behavior population" (mediating mounting and USVs in females and males) rather than a set of neurons specific to social isolation.I strongly recommend including experimental groups that involve labeling neurons after 30minute social interactions in group-housed female or male mice and inhibit TRAPed neurons after social isolation or activate TRAPed neurons after group housing. If manipulating the grouphoused TRAP neurons has similar effects to manipulating the isolated TRAP neurons, it would suggest the current labeling paradigm is not isolating neurons specific to the effect of social isolation per se. Rather, the neurons may mediate more general social interaction or motivationrelated activities. Given the known role of POA in male mating behavior, a group-housed TRAP experiment in males with a female visitor is especially important for understanding the selectivity of the labeled cells.Without proper controls, referring to the labeled neurons as "POAiso" neurons is potentially misleading. The data thus far suggests these neurons may predominantly reflect a "POA social behavior" population rather than a set of cells distinctly responsive to isolated housing.

We agree with the reviewer that the POA neurons we are studying regulate the production of social behaviors in females and males, rather than representing a set of cells distinctly responsive to single housing. To more clearly reflect our thinking, we have changed the name of the neurons from “POA-iso neurons” to “POA-social neurons”. Thank you for this helpful criticism.

Our Fos data are consistent with the idea that the POA may regulate social behaviors in group-housed females (not just single-housed females). Namely, we found that counts of Fospositive POA neurons are significantly related to rates of social investigation (p = 0.01) and tend to be related to USV rates (p = 0.05) in group-housed females that engaged in same-sex interactions (Fig. S1C). We now include two new sets of experiments aimed at further testing the idea this idea.

First, we include 2 control groups in which TRAPing sessions were performed in *grouphoused females* following same-sex interactions. We find that chemogenetic silencing of grouphoused-TRAPed POA neurons fails to reduce social behaviors in females that are subsequently single-housed and given a same-sex social interaction (Fig. 5A-D), and that optogenetic activation of group-housed-TRAPed POA neurons fails to promote female social behavior (Fig. 5E-H). At face value, these findings do not support the idea that the POA contains neurons that regulate social behaviors in group-housed females.

However, one important caveat is that group-housed females engage in low rates of social behaviors (low investigation time, no mounting, and few USVs), and thus TRAP-based labeling may not work efficaciously in these mice. There may be POA neurons that regulate social behaviors in group-housed females but that do not upregulate Fos following production of relatively low rates of social behaviors. To test this idea, we also include females in which POA neurons are chemogenetically silenced using a viral strategy that does not depend on activitydependent labeling. In this new experiment, we report that silencing of POA neurons significantly reduces USV production in group-housed females (Fig. 5J-L) and significantly reduces social investigation, mounting, and USV production when these same females are retested following single-housing (Fig. 5M-O). Together, these experiments suggest that the POA may regulate the production of social behaviors during same-sex interactions in group-housed females, but that these effects may be difficult to detect in some cases given the low rates at which group-housed females engage in social behaviors during same-sex interactions relative to single-housed females.

Finally, we want to highlight an additional new dataset that supports the idea that POAsocial neurons regulate social behaviors, rather than encoding the “state” of social isolation. We now include a control group for the chemogenetic silencing of female POA-social neurons, in which females were single-housed but were not given a social interaction prior to 4-OHT treatment (N = 5 non-social controls). Rates of social behaviors were subsequently unaffected following CNO delivery in these females (Fig. S2D-G). These new data support the conclusion that POA-social neurons regulate the production of social behaviors, rather than encoding the state of social isolation.

**Reviewer 3:**
While the authors should be commended for performing and reporting multiple circuit perturbation experiments (e.g., chemogenetics, ablation), the conflicting effects on behavior are hard to interpret without additional experiments. For example, chemogenetic silencing of the POA neurons (using DREADDs) attenuated all three behavioral measures but the ablation of the same POA neurons (using CASPACE) decreased mounting duration without impacting social investigation or USV production. Similarly, optogenetic activation of POA neurons was sufficient to generate USV production as reported in earlier studies but mounting or social investigation remained unaffected.Do these discrepancies arise due to the efficiency differences between DREADD-mediated silencing vs. Casp3 ablation? Or does the chemogenetic result reflect off-manifold effects on downstream circuitry whereas a more permanent ablation strategy allows other brain regions to compensate due to redundancy? It is important to resolve whether these arise due to technical reasons or whether these reflect the underlying (perhaps messy) logic of neural circuitry. Therefore, while it is clear that POA neurons likely contribute to multiple behavioral readouts of social isolation, understanding their exact roles in any greater detail will require further experiments.

We have added new analyses to consider the possibility that optogenetic activation of female POA-social neurons promotes social investigation. In the original manuscript, we analyzed the duration of social investigation bouts in POA-social-ChR2 females according to whether they overlapped with laser stimulation or whether they did not overlap. We realized that we made an error in this first analysis and inadvertently included social investigation bouts that occurred during the first 5 minutes of the social sessions, prior to any laser stimulation. Because these earlier bouts tend to be longer duration than later bouts, this mistake washed out the effect of laser stimulation on social bout duration. After correcting that error, we now report that optogenetic activation of female POA-social neurons lengthens social investigation bout duration (Fig. 4G). Inspired by this interesting finding, we also included analyses of the probability of social investigation following laser stimulation (Fig. 4E-F; excluding laser stimulations that were preceded by social investigation in the pre-laser baseline period). These analyses support the conclusion that optogenetic activation of POA-social neurons promotes both USV production and social investigation in group-housed females.

The majority of the females that we used in our TRAP2-based ablation experiments were heterozygous for TRAP2 (N = 11 of 15 POA-social-caspase subjects were TRAP2;Ai14 females), whereas all females used in our chemogenetic silencing experiments were homozygous for TRAP2. To test whether a more effective ablation of POA-social neurons might drive decreases in social investigation and USV production, we set up additional TRAP2 homozygous POA-social-caspase females and directly compare the effects of ablation between the two genotypes (Fig. S3; N = 11 hets in total and N = 9 homozygotes in total). These experiments revealed that effects on mounting were more pronounced following POA-social ablation in TRAP2 homozygotes vs. heterozygotes, but that neither group exhibited decreased social investigation or USV production following 4-OHT treatment.

To ask whether caspase-mediated ablation in TRAP2 homozygotes was effective in eliminating neural activity associated with social behaviors in females, we performed Fos immunostaining in a subset of the POA-social-caspase TRAP2 homozygotes following a samesex interaction. We found that POA Fos expression was robustly reduced in these females relative to control group-housed and control single-housed females that also engaged in samesex interactions, down to levels seen in group-housed and single-housed females that did not engage in a social interaction (comparison shown in Fig. S3D; control female data same as in Fig. 1). Moreover, the remaining POA Fos in these TRAP2 homozygotes was no longer positively correlated to social investigation or USV production (Fig. S3E-F). Together, these findings lead us to favor the interpretation suggested by the reviewer below, that permanent ablation of POA-social neurons leads to compensation from other brain regions due to redundancy. In addition, our finding that optogenetic activation of POA-social neurons promotes both USV production and social investigation supports the idea that POA-social neurons directly regulate these behaviors. We agree with the reviewer that additional work is needed to understand the complex sex- and context-dependent role played by the POA in the regulation of mouse social behaviors.

**Recommendations for the Authors:**

**Reviewer 1 Recommendations:**
(1) The largest issue is that many of the stated "key" behavioral findings are not statistically significant.(1a) Figure 2C is not significant and Figure 5G is not significant

We have added N = 5 POA-social-hM4Di females, N = 3 POA-social-hM4Di males, and N = 3 POA-social-GFP males to the dataset. The decrease in mounting following chemogenetic silencing of POA-social neurons is now statistically significant in both sexes (p < 0.05 for both; see current Figs. 2C and 6G). We also simplified our statistical analysis of mounting in these experiments to consider the proportion of trials with and without resident-initiated mounting on saline vs. CNO days, using McNemar’s test for paired proportions.

(1b) Mounting graphs are completely omitted in Figure 4.

Given that mounting was only observed infrequently in POA-social-ChR2 females, we simply report this information in the Results text (lines 382-388). In our prior summary of the mounting results, we reported that mounting was observed in a total of 3 trials from 2 females, but we inadvertently included information from a duplicate trial from one of the POA-socialChR2 females in this summary (all other analyses of the POA-social-ChR2 females included one trial per female). We have corrected that error and now report that we observed mounting following laser stimulation in 1 trial from 1 POA-social-ChR2 female. We have expanded our consideration of potential effects of optogenetic activation of POA-social neurons on social investigation and include these new analyses as part of Figure 4 (Fig. 4E-G), following the existing analyses of USV production.

(1c) Figure 3C shows a reduction of mounting following the ablation of POA (although no stats on the graph to denote significance), but this ablation approach can't resolve whether POA is required to encode the state produced by the short period of isolation, and/or whether it needs to be online at test.

We have now added an asterisk in Fig. 3C to denote a p value less than 0.05. Thank you for catching our oversight.

We designed our activity-dependent labeling experiments to TRAP and express viruses in POA neurons that increase their activity in conjunction with the production of social behaviors in single-housed females. We believe our findings our most consistent with the conclusion that these neurons regulate the production of social behaviors, rather than encoding the state of social isolation, and we have renamed these neurons as “POA-social” neurons to better reflect our thinking.

We also now include control experiments (albeit chemogenetic inhibition, not caspase ablation) in which the TRAP2 strategy is used to express hM4Di in the POA of single-housed females that do not experience a social interaction prior to 4-OHT delivery (non-social controls, Fig. S2D-G). We report that chemogenetic inhibition of these neurons does not decrease social behavior in single-housed females during a subsequent same-sex interaction (p > 0.05 for saline vs. CNO rates of social investigation, mounting, and USVs). These additional findings support the idea that the activity of POA-social neurons is related to the production of social behaviors rather than to the state of social isolation.

The reviewer is correct that our ablation approach cannot resolve the question of whether POA-social neuronal activity is required online during testing, but our reversible chemogenetic inhibition experiments provide evidence that the activity of POA-social neurons is required online at the time of testing to regulate social behavior.

(1d) A similar issue is seen regarding investigation (a general lack of significance with most of the LOF and GOF manipulations).

As reported in the original manuscript, we find that chemogenetic inhibition of POAsocial neurons reduces social investigation in females, while caspase-mediated ablation of female POA-social neurons does not. Our original caspase dataset used mostly but not all TRAP2 heterozygous females (N = 11 TRAP2 heterozygotes (TRAP2;Ai14), generated by crossing TRAP2 mice with Ai14 mice, for the purpose of visualizing the absence of tdTomato labeling to estimate spread of the caspase virus; and N = 4 TRAP2 homozygotes). By adding to the TRAP2 homozygous caspase dataset and comparing the effects on female social behavior of ablation of POA-social neurons in TRAP2 heterozygous vs. TRAP2 homozygous females, we

now provide evidence that the attenuation of mounting is more efficacious in TRAP2 homozygous females than in heterozygotes (Fig. S3B). Nonetheless, we fail to see effects on social investigation and USV production, even when caspase ablation of POA-social neurons is performed in TRAP2 homozygous females (Fig. S3A,C).

In spite of the lack of effect on these behaviors, we show that caspase-mediated ablation of POA-social neurons in TRAP2 homozygous females leads to a dramatic reduction in social interaction-induced Fos expression in the POA. POA Fos expression in these caspase females is reduced to the levels seen in control group-housed and single-housed females that are not given social interactions and are significantly lower than Fos expression in group-housed and single-housed females that are given a same-sex interaction (Fig. S3D). Moreover, the remaining POA Fos expression in the caspase females is no longer related to rates of social investigation (Fig. S3E), as is normally the case in group-housed and single-housed control females (Fig. S1C, left). Together, these data support the idea that some type of neuronal compensation outside of the POA is occurring following ablation of POA-social neurons, and this compensation permits normal levels of USV production and social investigation.

As in the original manuscript, we report that chemogenetic inhibition of POA-social neurons in male mice reduces mounting but does not reduce social investigation (or USV production). We now include quantification of social behaviors produced by male and female POA-social-hM4Di mice in the TRAPing sessions that preceded 4-OHT delivery (Fig. S5). These measurements show that males spent significantly more time than females engaged in mounting, and we speculate that this bias in TRAPing session behavior might have led to a bias in TRAP-mediated viral labeling of male POA neurons that regulate mounting, at the expense of male POA neurons that regulate social investigation (or USV production).

We have added new analyses to consider the possibility that optogenetic activation of female POA-social neurons promotes social investigation. In the original manuscript, we analyzed the duration of social investigation bouts in POA-social-ChR2 females according to whether they overlapped with laser stimulation or whether they did not overlap. We realized that we made an error in this first analysis and inadvertently included social investigation bouts that occurred during the first 5 minutes of the social sessions, prior to any laser stimulation. Because these earlier bouts tend to be longer duration than later bouts, this mistake washed out the effect of laser stimulation on social bout duration. After correcting that error, we now report that optogenetic activation of female POA-social neurons lengthens social investigation bout duration (Fig. 4G). Inspired by this encouraging finding, we also included analyses of the probability of social investigation following laser stimulation (Fig. 4E-F; excluding laser stimulations that were preceded by social investigation in the pre-laser baseline period). These analyses support the conclusion that optogenetic activation of POA-social neurons promotes both USV production and social investigation in group-housed females.

*(2) In Figure 1 and elsewhere, the authors use a Mann-Whitney U test, which should be used for non-parametric data, but in other places, they use statistical tests for normally distributed data. Why? How was the normality of distributions tested?*

We tested the normality of data distributions using the Shapiro-Wilk test. Parametric tests were used for analyses that contained normally distributed data, and non-parametric tests were used for analyses that contained non-normally distributed data. This information is included in the Methods (lines 997-1000), and full details of statistical analyses can be found in Table S1.

(3) The method for "trapping" neurons that are part of the short-term isolation ensemble has some caveats that have not been adequately addressed. First, 4-OHT was administered after social interaction, but before 24 hours of isolation, making it unclear exactly WHAT is being trapped.i) Is it neurons that encode the recent 3-day iso experience? (seems unlikely, as this would have been hours after the end of that iso window)

We now include a group of control females to directly test this possibility (Fig. S2D-G). These TRAP2 females were single-housed for 3 days but were not given a social interaction prior to 4-OHT treatment (N = 5 non-social controls). Presumably, POA neurons TRAPed in these females might encode the experience of short-term isolation. However, we found that chemogenetic inactivation of these TRAPed neurons during a subsequent same-sex interaction failed to decrease social behaviors in single-housed females (Fig. S2E-G; p > 0.05 for CNO vs. saline rates of social investigation, mounting, and USV production). These control experiments support the idea that we are TRAPing neurons whose activity is related to the production of social behaviors, and we have renamed the neurons as “POA-social” neurons to reflect this thinking.

ii) Is it neurons that encode the recent behavior impacted by the 3-day iso? (this seems to be the goal, but the authors do not provide evidence that the time course of their injection is efficient enough to recruit the recently activated neurons, nor do they provide evidence that opening the trapping window directly after the behavior is better than directly before)

We opted to perform IP injections of 4-OHT immediately following the behavior session, rather than behavior, due to concern that handling the mice and delivering IP injections prior to behavior sessions would stress the mice, leading to lower rates of social behaviors. The nonsocial female hM4Di experiments described above support the idea that we are TRAPing neurons related to the production of social behaviors, as the reviewer suggests.

iii) Is it trapping neurons active during the subsequent 24 hours of isolation? (seems possible, but this would mean that the authors are looking at a different population of neurons than they claim).

If chemogenetic silencing of POA neurons that were TRAPed following 3-days of social isolation but in the absence of a social interaction (N = 5 non-social controls, Fig. S2D-G) does not alter social behaviors, there is no compelling reason to hypothesize that TRAPing POA neurons activated following the 24 hours of social isolation that follow a social interaction would do so. Moreover, in the original study characterizing the TRAP2 mice (DeNardo et al., 2019), the authors performed experiments to characterize the time course of TRAPing relative to 4-OHT treatment and concluded that the majority of TRAPing occurs within a 6-hour window centered around the 4-OHT injection.

(4) Relatedly, the authors seem to find a fair bit of variability in their TRAP-mediated experiments. This begs the question - are the effects of their GOF and LOF approachesi) dependent on the iso-behaviors that were "trapped" for each animal (in other words, how does behavior at test 1 correlate with behavior at test 2)?

To test the reviewer’s idea, we compared rates of TRAPing session behaviors for the POA-social-hM4Di females to the subsequent effects of neuronal silencing on these behaviors calculated as (CNO behavior – saline behavior). These correlations are shown in Fig. S2A-C and are all non-significant. We also include below for the reviewer the same types of correlations for the other datasets in our study (loss-of-function experiments: female POAsocial-caspase, male POA-social-hM4Di; and gain-of-function experiments: female POA-socialChR2).

**Author response image 1. sa4fig1:** 

The only loss-of-function experiment comparison in the above figure that reveals a negative *and* significant correlation is the mounting comparison for the POA-social-hM4Di males (time spent mounting during TRAPing session vs. CNO time spent mounting -saline time spent mounting). This significant correlation likely reflects that fact that (1) no males mounted in the CNO session and (2) that mounting rates for individual males are relatively consistent over time (in comparison to female mounting, which is more variable; see Author response image 2 below of TRAPing session vs. saline mounting in male vs. female POA-social-hM4Di experiments). The correlation between TRAPing session and testing session mounting is significant for the POA-social-ChR2 females, but despite the significant correlation, we would want to see more instances of optogenetically-elicited mounting to make any claim about its relationship to TRAPing session behavior.

**Author response image 2. sa4fig2:** 

Nonetheless, we agree with the reviewer’s intuition that one would expect the effects of POA activity manipulations on different behaviors to scale with rates at which these behaviors were performed during the TRAPing session. We speculate that variability in the TRAPing process might have obscured such a relationship. There is inevitable variability in the exact body cavity placement of IP injections, which can affect drug absorption, and another point is that we delivered a fixed volume of 4-OHT (10 mg/mL 4-OHT in 150 uL filtered corn oil) to all mice in the study, regardless of their weight, which likely added variability in TRAPing efficacy from animal to animal. This detail was reported inaccurately in the Methods, and that error has been corrected (line 920). With regard to our male POA-social-hM4Di dataset, we find that these males spend more time mounting during their TRAPing sessions than female POA-socialhM4Di (Fig. S5; males also spent less time investigating and tended to produce fewer USVs than females), a fact that we hypothesize may have led to a bias toward TRAPing mountingrelated POA neurons in male subjects. In addition, however, the fact that male mice typically weigh more than females and would have received a slightly lower effective dosage of 4-OHT may also have contributed to the weaker effects on behavior in the male POA-social-hM4Di experiments relative to the female POA-social-hM4Di experiments.

We also want to highlight that interpreting correlations for females between time spent mounting during the TRAPing session and time spent mounting during the test sessions can be complicated. For example, we see 2 cases in the female POA-social-hM4Di dataset in which the female did not mount in the TRAPing session, and then mounted on the saline day (12s and 10s total mounting for those 2 females) but not on the CNO day. One interpretation of the data from these 2 females is that mounting on the TRAPing day is not required to attenuate mounting on the later test days. However, female mounting behavior itself is variable, both across different females and across different tests of a given female, as noted above. If we consider all singlehoused females included in our dataset for which we quantified control behavioral data (i.e., behavior trials from unmanipulated females and TRAPing sessions from females that were later manipulated), we find that mounting is not observed in ~30% of the females (24 of 83). In ongoing behavioral experiments not included in this manuscript, we are investigating factors that regulate female mounting following single-housing. In that dataset, we also see little evidence that female mounting in one social interaction predicts mounting in a subsequent interaction

(i.e., there don’t appear to stable “high mounters” and “low mounters” following single housing). Thus, the small number of cases in which females did not mount in the TRAPing session and then displayed mounting on the CNO only day are difficult to interpret.

Two additional considerations are that TRAPing may not be equally efficacious for POA neurons that regulate different behaviors, and that different behaviors may be differentially sensitive to perturbations of the POA. Previous elegant calcium imaging work has shown that different subsets of Esr1+ POA neurons exhibit activity that is “tuned” to specific behaviors (sniffing vs. mounting in males interacting with females; Yang et al., 2023). However, it is possible that these subsets of neurons display differential levels of Fos expression following the production of their preferred behavior and that some behavior-related subsets may thus be more easily TRAPed than others. It may also be the case that some behaviors are more easily disrupted by POA activity manipulations than others (e.g., perturbation in a smaller percentage of behavior-related POA neurons may be required to disrupt some behaviors relative to others).

Despite these caveats, we have two lines of evidence that the effects of chemogenetic silencing of POA-social neurons depends on the behaviors produced during the TRAPing sessions.

(1) Social behavior is required during the TRAPing session to see subsequent effects on social behavior following chemogenetic silencing of TRAPed POA neurons. In control females that were single-housed but were not given a social interaction prior to 4OHT treatment, social behaviors are not reduced by chemogenetic silencing of TRAPed POA neurons (Figs. S2D-G).

(2) To directly test whether mounting in the TRAPing session is required to see attenuation of mounting during subsequent chemogenetic silencing of POA-social neurons, we performed control experiments in which single-housed females interacted with a female visitor that was placed under a cup during the TRAPing session prior to 4-OHT treatment. Mounting was not possible in this context, and we also found that females produced lower rates of USVs during the TRAPing session relative to single-housed females engaged in free social interaction. However, subject females spent more time engaged in social investigation of the visitor relative to single-housed females engaged in free social interactions (see Author response image 3 below).

**Author response image 3. sa4fig3:** 

Unfortunately, none of the experimental females in this cohort displayed mounting in the CNO or saline sessions. Given that we could use this dataset to address the intended question, we did not include it in the manuscript. However, it is quite interesting that female subjects displayed higher than normal social investigation and lower than normal USV production in their TRAPing sessions (relative to single-housed females engaged in free interactions), and subsequently, chemogenetic inhibition of TRAPed POA neurons decreased social investigation but did not decrease USV production (Author response image 4 below).

**Author response image 4. sa4fig4:** 

Together, we think our data support the idea that the POA neurons that are TRAPed are related to the social behaviors performed by the animals, but these relationships may be complex and difficult to detect from comparisons across animals within a single experimental group.

And/or are theyii) influenced by the spread or amount of virus for each animal? These correlations could help shed light on what exactly is being trapped - is it specific behaviors or is it the "state" of shortterm isolation?

Our control experiments with females that were single-housed but did not receive a social interaction prior to 4-OHT treatment provide evidence that the production of social behaviors is required to see subsequent effects on behavior following chemogenetic inhibition of TRAPed POA neurons (Figs. S2D-G).

The same volume of virus was injected across all activity manipulation experiments (200 nL). Because of the trajectory of our POA viral injections (performed at a slight rostral angle relative to vertical), we did sometimes see viral labeling that spread into the AH caudal to the POA. For this reason, we included the AH TRAPed control group (Fig. 2), to rule out the possibility that viral spread into the AH could account for the effects of chemogenetic silencing of POA-social neurons on female social behaviors. Also because of the injection angle used, we don’t see substantial viral spread rostral to our injection coordinates. In short, there isn’t systematic variability in the targeting or spread of our POA viral injections that can account for variability in the effects on USV production and social investigation of our LOF and GOF manipulations (female hM4Di and female ChR2 experiments).

In older lesion studies in male rodents and birds, there is some support for the idea that rostral vs. caudal POA neurons differentially regulate appetitive vs. consummatory sexual behaviors (as reviewed in Balthazart and Ball, 2007). However, all of our viral injections were placed in what that review paper would have considered ‘caudal’ POA. We also note that more recent imaging studies have reported that subsets of POA neurons are differentially tuned to male sniffing vs. male mounting (Yang et al.,2023), and these subsets must be relatively co-localized given that they are imaged in the same field of view. Whether distinct subsets of POA neurons regulate the production of different female social behaviors, and if so, how these subsets are localized within the POA, remains an important question for future study.

(5) The authors label their region of interest as the "POA" but images throughout (e.g. their fos image, Figure 1E), look more like the MPO. Why label it POA?

The POA neurons in our study are found in a band that spans the medial POA, as well as a bit of the lateral POA. To avoid over-specifying, we call this region the POA more generally.

(6) In all the experiments, mice are isolated and then re-group housed with siblings. Do all the siblings in the group belong to the same experimental group, or are siblings naïve? This may be critical to help determine whether some of the effects observed may be "group" effects.

In general, multiple (although not always all) mice in a cage belonged to the same experimental group. In our inhibitory DREADDs experiments, it is unclear how that could drive our observed effects on behavior, given that home cage behavior would only be expected to differ for a given mouse in the time period following their CNO session.

For the female POA-social-caspase mice, we cannot rule out the possibility that their home cage behaviors differed in the time period following 4-OHT treatment and re-grouphousing and prior to post-4-OHT behavior measurements. However, given that the only social behavior affected by ablation of POA-social neurons was mounting, and that rates of mounting would be expected to be very low in group-housed females within home cages, it is unclear how our experimental result could be attributed to group effects.

If by “group” effects the reviewer means “litter” effects, we include a plot below that shows the CNO vs. saline behaviors for the POA-social-hM4Di females, separated by cage ID. There is no evidence that the effects of chemogenetic silencing of POA-social-hM4Di females are being driven by only certain cages (only social investigation and USVs are shown, because mounting was uniformly low (1 of 17 females mounted) in the CNO session).

**Author response image 5. sa4fig5:** 

(7) For chemogenetic experiments, the authors state that CNO and Saline were given in a counterbalanced order (eg line 189). Did the authors see any order effects?

We did not see order effects, and we can include plots of those data below for the female and male POA-social-hM4Di groups, with mice plotted according to which treatment they received first.

**Author response image 6. sa4fig6:** 

(8) In the control experiments in Figure 2 where VMH or AH are chemogenetically silenced, it isn't clear whether these groups include mice that were subjected to 3 days of isolation. Please clarify.

Yes, these female groups were also subjected to 3 days of isolation (first prior to the TRAPing session, and for a second time prior to the onset of the CNO/saline testing sessions). That information has been clarified in the Results section (line 214) and in the Methods (lines 935-938).

(9) Line 312. The title for this section, "POA neurons increase their activity....." is somewhat misleading. It sounds like the authors imaged trapped neurons. I think what they mean is that more POA neurons are activated following opposite-sex interactions with males.

Thanks for this catch. We have modified the section title, as well as the title of the first results sub-section.

(10) Figure 5A, right panels. The authors fail to find an increase in the investigation of male-male pairs following the short-term isolation of one. This contrasts with the main finding in Matthews et al., 2016 Cell, where short periods of isolation are said to promote pro-social behaviors. The authors could comment on this discrepancy in their discussion (eg difference in testing apparatus/test type? Difference in the number of days of isolation? etc.).

In current Fig. 6A, there is no significant interaction between the two main effects, but each main effect is significant: single-housed males spend more time investigating partners than group-housed males, and males spend more time investigating female partners than male partners. The significant main effect of housing condition is consistent with the findings of Matthews et al., 2016 and is included within the Results (lines 486-492).

(11) Figure 5F, the authors seem to have a main effect of virus (more overall investigation in dreadds mice). Nothing about this is addressed.

We sometimes see differences in social behavior between cohorts of males when they are tested at different times and, correspondingly, with different groups of female social partners. Our POA-social-hM4Di and POA-social-GFP males were set-up and tested at largely non-overlapping times. We have added a brief note to the Results section to include this information (lines 535-539).

**Reviewer 2 Recommendations:**
(1) (C)ritical control experiments are missing to support this claim (that a population of preoptic hypothalamic neurons contribute to the effects of short-term social isolation on the social behaviors of female mice).(1a) All the activity-dependent labeling experiments with TRAP mice, including the subsequent neural activity manipulation experiments (Figures 2, 3, 4, 5E-F), were conducted by labeling neurons only in socially isolated animals, not group-housed animals. The authors labeled neurons after 30-minute social interactions, raising the possibility that the labeled neurons simply represent a "social interaction/behavior population" (mediating mounting and USVs in females and males) rather than a set of neurons specific to social isolation behaviors of mice… The data thus far suggests these neurons may predominantly reflect a "POA social behavior" population rather than a set of cells distinctly responsive to isolated housing.

We agree with the reviewer that the POA neurons we are studying regulate the production of social behaviors in females and males, rather than representing a set of cells distinctly responsive to single housing. To more clearly reflect our thinking, we have changed the name of the neurons from “POA-iso neurons” to “POA-social neurons”. Thank you for this helpful criticism.

Our Fos data are consistent with the idea that the POA may regulate social behaviors in group-housed females (not just single-housed females). Namely, we found that counts of Fospositive POA neurons are significantly related to rates of social investigation (p = 0.01) and tend to be related to USV rates (p = 0.05) in group-housed females that engaged in same-sex interactions (Fig. S1C). We now include two new sets of experiments aimed at further testing the idea this idea.

First, we include 2 control groups in which TRAPing sessions were performed in *grouphoused females* following same-sex interactions. We find that chemogenetic silencing of these group-housed-TRAPed POA neurons fails to reduce social behaviors in females that are subsequently single-housed and given a same-sex social interaction (Fig. 5A-D; GH-TRAPed POA hM4Di females), and that optogenetic activation of group-housed-TRAPed POA neurons fails to promote female social behavior (Fig. 5E-H; GH-TRAPed POA ChR2 females). At face value, these findings do not support the idea that the POA contains neurons that regulate social behaviors in group-housed females.

However, one important caveat is that group-housed females engage in low rates of social behaviors (low investigation time, no mounting, and few USVs), and thus TRAP-based labeling may not work efficaciously in these mice. There may be POA neurons that regulate social behaviors in group-housed females but that do not upregulate Fos following production of relatively low rates of social behaviors. To test this idea, we also include females in which POA neurons are chemogenetically silenced using a viral strategy that does not depend on activitydependent labeling. In this new experiment, we report that silencing of POA neurons significantly reduces USV production in group-housed females (Fig. 5J-L) and significantly reduces social investigation, mounting, and USV production when these same females are retested following single-housing (Fig. 5M-O).

(2) Please add strain background information of subject animals in the methods.

This information has been added to the Animals section within the Methods (lines 788802).

**Responses to Reviewer 3 Recommendations:**
(1a) (T)he conflicting effects on behavior are hard to interpret without additional experiments….Similarly, optogenetic activation of POA neurons was sufficient to generate USV production as reported in earlier studies but mounting or social investigation remained unaffected.

We have added new analyses to consider the possibility that optogenetic activation of female POA-social neurons promotes social investigation. In the original manuscript, we analyzed the duration of social investigation bouts in POA-social-ChR2 females according to whether they overlapped with laser stimulation or whether they did not overlap. We realized that we made an error in this first analysis and inadvertently included social investigation bouts that occurred during the first 5 minutes of the social sessions, prior to any laser stimulation. Because these earlier bouts tend to be longer duration than later bouts, this mistake washed out the effect of laser stimulation on social bout duration. After correcting that error, we now report that optogenetic activation of female POA-social neurons lengthens social investigation bout duration (Fig. 4G). Inspired by this interesting finding, we also included analyses of the probability of social investigation following laser stimulation (Fig. 4E-F; excluding laser stimulations that were preceded by social investigation in the pre-laser baseline period). These analyses support the conclusion that optogenetic activation of POA-social neurons promotes both USV production and social investigation in group-housed females.

(1b) Do these discrepancies (between hM4Di and caspase) arise due to the efficiency differences between DREADD-mediated silencing vs. Casp3 ablation? Or does the chemogenetic result reflect off-manifold effects on downstream circuitry whereas a more permanent ablation strategy allows other brain regions to compensate due to redundancy? It is important to resolve whether these arise due to technical reasons or whether these reflect the underlying (perhaps messy) logic of neural circuitry.

The possibility that the difference in effects on behavior between chemogenetic silencing and caspase ablation at face value seems inconsistent with the findings of previous experiments, in which ablation of large numbers of POA neurons failed to reduce USV production in male mice (POA lesions in Bean et al., 1981; ablation of VGAT+ POA neurons by Gao et al., 2018). These findings stand in contrast to those using chemogenetic silencing of large numbers of POA neurons, which report reduced USV production in male mice (VGAT+/Esr1+ in Karigo et al., 2021; Esr1+ in Chen et al., 2021).

However, it is the case that the majority of the females that we used in our TRAP2-based ablation experiments were heterozygous for TRAP2 (N = 11 of 15 POA-social-caspase subjects were TRAP2;Ai14 females), whereas all females used in our chemogenetic silencing experiments were homozygous for TRAP2. To test whether a more effective ablation of POAsocial neurons might drive decreases in social investigation and USV production, we set up additional TRAP2 homozygous POA-social-caspase females and directly compare the effects of ablation between the two genotypes (Fig. S3; N = 11 hets in total and N = 9 homozygotes in total). These experiments revealed that effects on mounting were more pronounced following POA-social ablation in TRAP2 homozygotes vs. heterozygotes, but that neither group exhibited decreased social investigation or USV production following 4-OHT treatment.

To ask whether caspase-mediated ablation in TRAP2 homozygotes was effective in eliminating neural activity associated with social behaviors in females, we performed Fos immunostaining in a subset of the POA-social-caspase TRAP2 homozygotes following a samesex interaction. We found that POA Fos expression was robustly reduced in these females relative to control group-housed and control single-housed females that also engaged in samesex interactions, down to levels seen in group-housed and single-housed females that did not engage in a social interaction (comparison shown in Fig. S3D; control female data same as in Fig. 1). Moreover, the remaining POA Fos in these TRAP2 homozygotes was no longer positively correlated to social investigation or USV production (Fig. S3E-F). Together, these findings lead us to favor the interpretation suggested by the reviewer below, that permanent ablation of POA-social neurons leads to compensation from other brain regions due to redundancy.

Given the negative results above, we favor this possibility and indicate so in our Discussion. In addition, our finding that optogenetic activation of POA-social neurons promotes both USV production and social investigation supports the idea that POA-social neurons directly regulate these behaviors. We agree with the reviewer that additional work is needed to understand the complex sex- and context-dependent role played by the POA in the regulation of mouse social behaviors.

(2) L 49: Please define Mesolimbic circuitry the first time it is mentioned.

We have added a definition (lines 52-53).

(3) L 210: In Figure 2C, the mounting duration baseline (saline) distribution seems lower than the same experimental baseline in Figures 1C and 3C. Does this reflect natural variability in the behavioral assay and might this be mitigated by additional sampling of animals?

Yes, there is substantial variability in the display of mounting behavior by single-housed females, including in the proportion of trials with mounting as well as in the total duration of mounting. In the revised manuscript, we have simplified our analysis of mounting in our TRAPbased experiments to quantify the proportion of trials with mounting, rather than considering the total time spent mounting. After adding N = 5 additional females to the POA-social-hM4Di dataset, we now report a statistically significant decrease in the proportion of trials with mounting following chemogenetic silencing of POA-social neurons (Fig. 2C; McNemar’s test for paired proportions).

(4) L 310: The authors claim that "These findings suggest that a subset of POAiso neurons overlap with GABAergic, PAG-projecting POA neurons that have been demonstrated in previous work to promote USVs via disinhibition of excitatory PAG neurons important to USV production (Chen et al., 2021; Michael et al., 2020)." I think the data reported suggests the opposite since only 18.3% of all POA->PAG neurons are cFos+. Perhaps better rephrased as "A subset (18.3%) of POA->PAG neurons are labelled by cFos and that is sufficient to drive the production of USVs". Is it surprising?

We modified the phrasing (lines 468-469), but a bit differently than suggested above, because although we suspect that optogenetic activation of the PAG-projecting neurons within the larger population of POA-social neurons is responsible for eliciting USV production, we did not technically demonstrate this to be the case in the current dataset.

We do find it surprising that so few (only ~20%) of PAG-projecting POA neurons upregulate Fos following female-female interactions marked by high rates of USV production. Even though optogenetic activation of PAG-projecting POA neurons elicits USV production, our finding suggests that the majority of PAG-projecting POA neurons may not play a role in regulating vocalization. In future work, it may be useful to apply an intersectional approach to further understand how the POA regulates USV production (for example, measure or manipulate activity selectively in projection-defined subsets of POA-social neurons).

(5) Given the considerable prior evidence of POA->PAG circuit in promoting USVs, it is hard to understand why chemogenetic inactivation of POA neurons in males affects mounting but not USV production (Figures 5F-H). Any potential explanation for this discrepancy?

We have two ideas about this surprising result. First, we examined the TRAPing session social behaviors of female and male POA-social-hM4Di mice. We found that male POA-socialhM4Di mice spent more time than female subjects mounting during the TRAPing sessions, and conversely, males spent less time investigating visitors and tended to produce fewer USVs than female subjects (Fig. S5). Given that our labeling method is activity-dependent, one possibility is that this bias in behavior is reflected in a bias toward labeling of POA neurons related to mounting.

Second, each mouse in the TRAP2-based hM4Di datasets received an IP injection of the same amount of 4-OHT (150 nL of 10 mg/mL 4-OHT in filtered corn oil) not adjusted for weight of the mouse. This information was not reported accurately in the Methods, and we have adjusted that section accordingly (line 920). As a result, because male mice typically weigh more than females and would have received a lower effective dosage of 4-OHT, another possibility is that TRAPing in males was less efficient than in females and accounts for the less complete effects on social behaviors. We have added language to the Results to discuss these possibilities (lines 540-560).

(6) L 472: Typo. "we found that short-term isolation exerts more robust on the effects of male behavior during subsequent interactions with females than during interactions with males."

Thank you for catching this mistake.